# Reducing Predictive Feature Suppression in Resource-Constrained Contrastive Image-Caption Retrieval

**Maurits Bleeker**  *m.j.r.bleeker@uva.nl*
*University of Amsterdam*

**Andrew Yates**  *a.c.yates@uva.nl*
*University of Amsterdam*

**Maarten de Rijke**  *m.derijke@uva.nl*
*University of Amsterdam*

**Reviewed on OpenReview:** *https://openreview.net/forum?id=T1XtOqrVKn*

## Abstract

To train image-caption retrieval (ICR) methods, contrastive loss functions are a common choice for optimization functions. Unfortunately, contrastive ICR methods are vulnerable to predictive feature suppression. Predictive features are features that correctly indicate the similarity between a query and a candidate item. However, in the presence of multiple predictive features during training, encoder models tend to suppress redundant predictive features, since these features are not needed to learn to discriminate between positive and negative pairs. While some predictive features are redundant during training, these features might be relevant during evaluation. We introduce an approach to reduce predictive feature suppression for resource-constrained ICR methods: *latent target decoding* (LTD). We add an additional decoder to the contrastive ICR framework, to reconstruct the input caption in a latent space of a general-purpose sentence encoder, which prevents the image and caption encoder from suppressing predictive features. We implement the LTD objective as an optimization constraint, to ensure that the reconstruction loss is below a bound value while primarily optimizing for the contrastive loss. Importantly, LTD does not depend on additional training data or expensive (hard) negative mining strategies. Our experiments show that, unlike reconstructing the input caption in the input space, LTD reduces predictive feature suppression, measured by obtaining higher recall@k, r-precision, and nDCG scores than a contrastive ICR baseline. Moreover, we show that LTD should be implemented as an optimization constraint instead of a dual optimization objective. Finally, we show that LTD can be used with different contrastive learning losses and a wide variety of resource-constrained ICR methods.

## 1 Introduction

Image-caption retrieval (ICR) is the task of using an image or a caption as a query and ranking a set of candidate items in the other modality. Both the images and captions are mapped into a shared latent space by two encoders, which correspond to the two modalities. These encoders usually do not share parameters and are typically optimized with a contrastive loss function (Faghri et al., 2018; Lee et al., 2018; Li et al., 2019; Wang et al., 2019; Messina et al., 2020b; Chen et al., 2020a; Liu et al., 2020; Wang et al., 2020; Messina et al., 2020a; Jia et al., 2021; Diao et al., 2021; Yu et al., 2021). How well an ICR method generalizes beyond the specific training data depends on the features that the method has learned during training. The contrastive loss explicitly learns the similarity between positive (matching) candidates, while pushing away negative (non-matching) candidates during training. In the ideal situation, the contrastive objective optimizes the image and caption encoder such that both encoders extract all relevant information from the

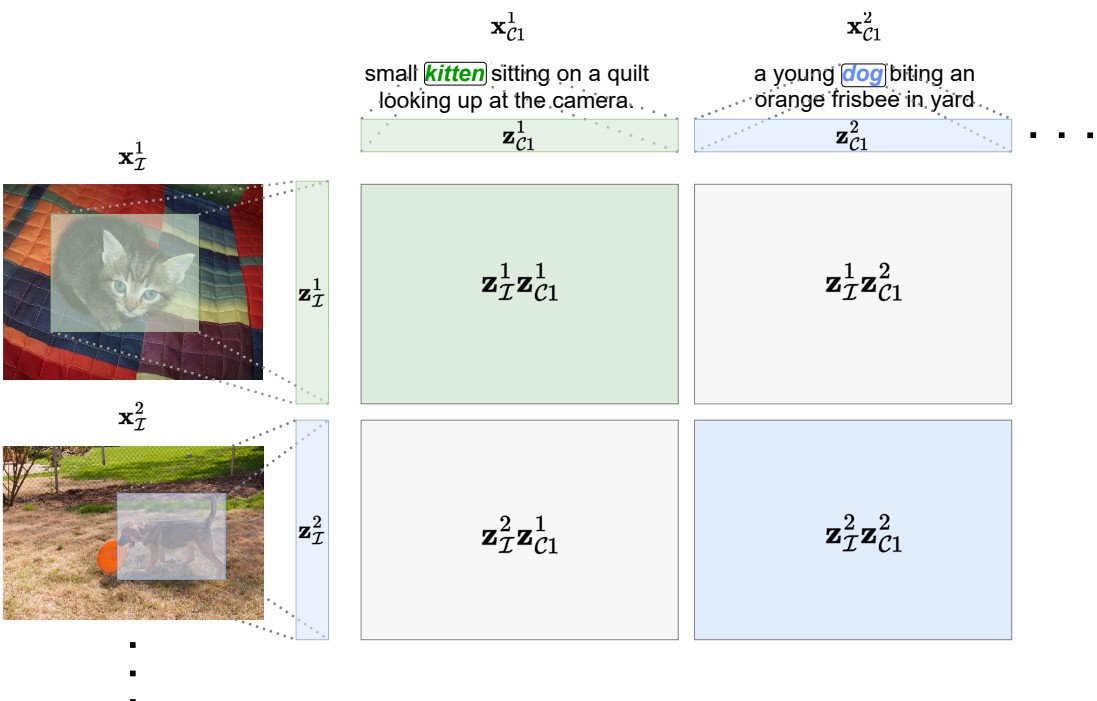

Figure 1: Visualization of *predictive feature suppression* using two examples from the MS-COCO captions (COCO) dataset. $\boldsymbol{x}_{\mathcal{I}}^*$ and $\boldsymbol{x}_{\mathcal{C}j}^*$ are input *images* and *captions*, respectively, and $\boldsymbol{z}_{\mathcal{I}}^*$ and $\boldsymbol{z}_{\mathcal{C}j}^*$ are the latent representations of the image and caption. We use cosine similarity as similarity metric. The objective of a contrastive loss is to optimize the similarity scores on the diagonal of the similarity matrix, while minimizing the off-diagonal scores. In this small-scale training setup, if both the image and caption encoder only extract the concepts **kitten** and **dog**, the remaining concepts in both the images and captions are irrelevant to predicting the correct matching scores between the images and captions. The input features that are not needed to predict a match between an image and a caption are likely to be *suppressed* by the encoder. However, these features might be relevant during evaluation to predict a correct match.

caption and image that is needed for matching the positive candidates during evaluation. However, it is not defined upfront what information is needed during evaluation for retrieving the correct item among a set of candidates.

**Predictive feature suppression.** Hermann & Lampinen (2020) show that, in the presence of two *predictive features* that redundantly predict the output label of the input data, a deep neural model preferentially represents one of the two predictive features while the other feature is suppressed. In this work, we define *predictive feature suppression* for ICR as the suppression of features by an encoder network during training that would be useful to correctly predict the match between a query and the positive candidate at inference time. For contrastive training tasks, the features that are relevant for matching the query with the positive candidate (i.e., the predictive features) mainly depend on the negative candidates in the training batch. Only optimizing the contrastive InfoNCE loss does not guarantee avoidance of *shortcut features* that suppress certain (predictive) input features and the learned features mainly depend on the difficulty of the discrimination task (Robinson et al., 2021). Especially in a resource-constrained training setup, it is likely that the majority of the input features in the caption and image are redundant for learning the similarity between matching images and captions, due to the limited number of negative samples available to contrast with. The contrastive optimization objective is easy to solve by only using a small subset of the predictive input features of the images and captions. Suppressing predictive features during training is an undesirable side-effect of contrastive representation learning in a resource-constrained training setup, since some of these features might be needed during evaluation to retrieve the matching candidate. In Figure 1, we provide a visual example of predictive feature suppression in a resource-constrained contrastive image-text matching setup.

**How to prevent predictive feature suppression.** To increase the difficulty of a contrastive discrimination task, one can increase the batch size during training in order to increase the probability of having difficult in-batch negative samples (Chen et al., 2020c; Qu et al., 2021). It is, therefore, not surprising that most progress on the two widely used ICR benchmark evaluation sets, Flickr30k (F30k) (Young et al., 2014) and MS-COCO captions (COCO) (Lin et al., 2014), has recently been made by using large-scale image-text matching training, mainly in combination with transformer network architectures (Jia et al., 2021; Yuan et al., 2021). Using more data and larger model architectures improves performance but comes with a significant extra computational cost, both in terms of data needed for training and the number of parameters that need to be optimized.

The two benchmark datasets for ICR, the F30k and COCO datasets, are relatively small in terms of training samples compared to the training data of state-of-the-art pre-trained ICR or image-text matching methods (Jia et al., 2021; Yuan et al., 2021). When an ICR method is trained from scratch using these benchmark datasets only, for example, in a resource-constrained training setup, scaling up the size of a batch is not a feasible solution to increase performance, due to the limited data size of F30k and COCO or due to the lack of computational resources. Hence, it is important to develop algorithms that can improve the effectiveness of ICR methods in a *resource-constrained* training setup, without relying on more data and more compute to achieve this.

A method to increase the difficulty of the contrastive objective that does not rely on the size of the dataset to reduce predictive feature suppression, is to directly mine *hard* negative examples for each query over the entire dataset, rather than relying on an increased batch size to include difficult negative examples. The disadvantage of hard-negative mining is that it can be computationally expensive (Chen et al., 2020b). Moreover, the COCO dataset contains many visually similar images (Parekh et al., 2020); when a similar image is mined as a hard-negative, it will be considered as a negative w.r.t. the query, which may create conflicting and incorrect supervision signals.

The autoencoding paradigm (Hinton & Salakhutdinov, 2006) provides an alternative solution to reduce predictive feature suppression by learning latent data representations that contain as much of the important input features as possible. Using the information bottle-neck principle, the encoder should compress the input information into a low-dimensional representation while preserving as much as possible of the input features. Combining autoencoding with contrastive learning should prevent the image and caption encoder from learning features that are only needed to solve the contrastive optimization objective. Therefore, a logical step is to add a decoder to the learning algorithm that decodes the original input from either the caption or image representation (or both). However, adding a decoder on top of the image representations, as in (Li et al., 2020a), is sub-optimal for the ICR task. The captions provided for each image are already a dense summary of the image; reconstructing every pixel in the image results in image representations that contain too much local information, which is irrelevant for the ICR task. A more natural choice would be to decode the input caption rather than the image, but adding a decoder on top of the caption representations might not result in a reduction of predictive feature suppression. Strong textual decoders can reduce a reconstruction loss by mainly relying on the learned language model (Lu et al., 2021). The input for this decoder (the latent caption representation) can mostly be ignored while correctly decoding the input sequence.

**Our proposed solution.** To address the disadvantages of current approaches to mitigating predictive feature suppression, viz. (i) high costs (in terms of compute and data), and (ii) reconstruction of the input caption and images in the input space, we introduce *latent target decoding* (LTD). For each caption in the training set, we generate a *latent target* representation by using a general-purpose sentence encoder. We train an image and caption encoder that can be trained in a resource-constrained setup, using a standard contrastive learning objective. Next to that, we add an extra decoder to the learning algorithm. We decode the information of the caption in a latent space of the sentence encoder. Thus, the decoder cannot rely on learning a dataset-specific language model to decode the input, and the caption representation learned by the caption encoder should contain all input features that are needed to decode the latent target. By reconstructing this latent target representation we aim to reduce predictive feature suppression by the caption encoder, which should result in representations that generalize better to the evaluation task. See Figure 2 in Section 3 for a high-level overview of our LTD method. LTD only requires an additional target representation for each caption and a simple feed-forward decoder network, and can be combined with any ICR method that

uses a separate caption and image encoder to compute a global representation of the input data. LTD does *not* depend on (i) additional training data, (ii) extra manual data annotation or (hard) negative mining, or (iii) significantly more computational resources. In this work, we focus on *resource-constrained* ICR methods that are trained from scratch on the F30k or COCO dataset on a single GPU.

If we were to add LTD to the learning algorithm, the overall training objective would become a multi-task loss: a contrastive and reconstruction loss. However, multi-task losses are difficult to optimize (Malkiel & Wolf, 2021). The reconstruction loss should serve as an extra regularizer rather than the main learning objective. We also do not want the caption encoder to mainly focus on the reconstruction objective, since that can harm the contrastive utility of the representations. Therefore, we implement LTD as an optimization constraint. In this manner, we can target a specific value for that loss function. The main training objective is to minimize the contrastive loss, given the constraint that the reconstruction loss is below a certain bound value. Similar to (Rezende & Viola, 2018; van Rozendaal et al., 2020), we implement the reconstruction loss constraint using a Lagrange multiplier (Platt & Barr, 1987); the two losses are scaled automatically such that the reconstruction bound is met, while minimizing the contrastive loss.

**Our main findings.**

1. The proposed constrained-based LTD reduces predictive feature suppression and improves the generalizability of learned representations, as it outperforms ICR baselines that are only optimized by using a contrastive loss. We measure the reduction of predictive feature suppression by using the standard evaluation metrics for the ICR task.
2. Implementing LTD as a dual loss, as opposed to an optimization constraint, does not reduce predictive feature suppression. Our analyses suggest that optimizing the reconstruction loss only until a specific bound value is met, results in better evaluation performance than minimizing the reconstruction loss as a dual loss.
3. LTD can be used in combination with different contrastive losses, for example, InfoNCE (van den Oord et al., 2018) and the triplet loss (Faghri et al., 2018), and it can be combined with a wide variety of ICR methods that can be optimized in a resource-constrained setup, such as VSRN (Li et al., 2019) and TERN (Messina et al., 2020b).

Below, we first cover related work, then introduce the proposed LTD method, before presenting our experimental setup and discussing the outcomes of our experiments, and concluding.

## 2 Related work

### 2.1 Image-caption retrieval

**Neural architectures for ICR.** We focus on ICR methods that compute a global representation for both image and caption. In general, an ICR method consists of two encoders: one to encode the image and one to encode the caption into a latent representation (Faghri et al., 2018; Li et al., 2019; Chun et al., 2021; Jia et al., 2021). Most work on ICR focuses on new network architectures to learn multi-modal feature representations. State-of-the-art results have been obtained using graph neural networks (Li et al., 2019; Wang et al., 2020; Liu et al., 2020; Diao et al., 2021) to represent visual relations in scenes as a graph, or attention mechanisms to align words in the caption with specific regions in the input image (Lee et al., 2018; Chen et al., 2019; Wang et al., 2019; Chen et al., 2020a; Yu et al., 2021; Zhang et al., 2022). Li et al. (2019) combine a caption encoder-decoder with the image encoder to add extra training signals to the learning algorithm. These methods are only trained and evaluated on the F30k and COCO datasets. Recently, there has been a shift to transformer-based (Vaswani et al., 2017) network architectures for both the image and caption encoder. Messina et al. (2020b;a) introduce a transformer-based network architecture solely trained for the ICR task. Since then, several transformer-based methods have been introduced (Lu et al., 2019; Chen et al., 2020d; Li et al., 2020b; 2021; Jia et al., 2021; Li et al., 2022); some combine the image and caption encoder into one unified architecture. These methods are all (pre-)trained on a large amount of additional training data and most are not trained for the ICR task specifically, but as general-purpose vision-text models.

**Hard negative mining.** Few publications have looked into the improvement of contrastive optimization for ICR methods. Faghri et al. (2018) introduce a new formulation of the triplet loss that only considers the hardest negative candidate in the training batch instead of all negative candidates, which significantly improved the evaluation scores on the ICR benchmarks. Since then, many ICR methods (Lee et al., 2018; Li et al., 2019; Wang et al., 2020; Liu et al., 2020; Messina et al., 2020b;a; Chen et al., 2020a; Diao et al., 2021; Yu et al., 2021) have used this loss function for optimization. Chen et al. (2020b) introduce an offline hard-negative mining approach for ICR in order to overcome the limitations of in-batch negative mining. Instead of mining an in-batch hard-negative, they mine additional negative candidates, for each query, over the entire training set to learn from so-called harder-to-distinguish negatives.

**One-to-many problem.** Chun et al. (2021) focus on the one-to-many problem in ICR. An image can be described by many captions. However, most methods in ICR learn one representation for the image, which should match with a number of different captions. They propose a probabilistic ICR method, where images and captions are represented as probability distributions in a shared latent space instead of a point representation. Although their method does not focus on contrastive optimization, it addresses predictive feature suppression by learning a distribution over features instead of a point prediction of features. Chun et al. (2021) also propose the plausible match metric, a heuristic for identifying missing positive examples by finding images that contain similar objects (i.e., plausible matches) and considering these in the evaluation.

Biten et al. (2022) propose semantic adaptive margin (SAM). Instead of using the binary relevance annotation between images and caption (of the F30k and COCO datasets) for the triplet loss computation, the authors propose an adaptive margin to model the many-to-many relation between images and caption. The standard triplet loss uses a fixed margin parameter $\alpha$. SAM dynamically assigns a unique distance value to the triplets in the training batch, based on the semantic similarity between an image and caption. In contrast, in this work we do not change the formulation of the contrastive loss. We add an extra optimization objective to the learning algorithm to prevent predictive feature suppression.

### Upshot

Unlike most previous work, we do not focus on the network architecture to improve the ICR performance. Similar to (Chun et al., 2021), we focus on small-scale learning set-ups to train an ICR method from scratch to show the strength of our method in a resource-constrained setting. Our proposed approach incorporates autoencoding into the learning algorithm in order to reconstruct the input caption to reduce predictive feature suppression.

### 2.2 Contrastive representation learning

Contrastive learning losses are used to learn discriminative representations of the input data that can be used to contrast positive and negative pairs of information in a latent space. These loss functions have made a big impact in representation learning, whether self-supervised (van den Oord et al., 2018; Chen et al., 2020c) or supervised (Radford et al., 2021; Karpukhin et al., 2020). Although ICR is a supervised contrastive learning task, some of the theoretical findings about self-supervised contrastive learning apply to supervised settings as well.

**Self-supervised contrastive learning.** A common approach to learn general-purpose representations in a self-supervised manner, is to create two (matching) views of the same (or similar) data point(s) by applying different augmentations (Chen et al., 2020c) or by splitting the data into parts (van den Oord et al., 2018) (i.e., predicting the future). The two positive views are contrasted with other negative samples in the training batch. The goal is to learn encoders that are invariant under these augmentations and that can discriminate between positive and negative pairs. How well self-supervised representations generalize to different settings, after training, is often assessed using a down-stream evaluation task, such as object classification (Chen et al., 2020c) or speaker identification (van den Oord et al., 2018).

Some work examines data augmentation to learn strong feature representations. Good augmentations retain task-relevant information while removing task-irrelevant nuisances (Tian et al., 2020). The main purpose of removing task-irrelevant nuisances is to prevent encoders from using this information as predictive features

during training. Xiao et al. (2021) show that the features needed to learn good representations depend on the down-stream task. ICR does not depend on augmentations to generate positive and negative pairs. These pairs are given by the annotations of the benchmark datasets (Young et al., 2014; Lin et al., 2014). The difficulty of the discrimination task (and hence the learned features) mainly depends on which candidates are present in the training batch.

The generalizability of contrastive learning methods is also influenced by the number of (hard) negatives present in a training batch. In general, the larger the number of in-batch negatives, the higher the down-stream evaluation performance (Chen et al., 2020c). Some work has focused on methods to increase the number of negatives during training (He et al., 2020) or on applying hard-negative mining strategies to increase the number of hard negatives in the batch (Lindgren et al., 2021; Xiong et al., 2021). Since we are focusing on a resource-constrained setup in this work, scaling up the batch size to increase the number of (hard) negatives is not a feasible solution. Moreover, the COCO dataset contains many visually similar images (Parekh et al., 2020). Mining visually similar images as (hard) negatives will result in a suboptimal supervision signal, which makes hard-negative mining also not a feasible approach to reduce shortcut feature suppression.

**Shortcut feature representations.** Robinson et al. (2021) show that the contrastive InfoNCE loss (van den Oord et al., 2018) does not guarantee avoidance of shortcut feature representations. Shortcut feature representations are solutions that suppress predictive input features, i.e., a shortcut to discriminate between matching/non-matching candidates. The features learned by the InfoNCE loss depend on the difficulty of instance discrimination during training. If the instance discrimination task is easy to solve during training, the model will learn shortcut features. Especially in a resource-constrained ICR training setup, the contrastive objective is easy to solve, since there is only a limited number of (hard) negative samples in the training batch, which will result in shortcuts/predictive feature suppression.

**Feature suppression among competing features.** Chen et al. (2021) introduce the notion of feature suppression among *competing features*. Chen et al. (2020c) show that, for example, the SimCLR method (Chen et al., 2020c) when trained without the crop or color augmentation (which randomly crops or shifts the color distribution of an image), shows a significant drop in performance on the down-stream evaluation task. Apparently, the (color) pixel distribution of the image is a sufficient predictive feature to match two views of the same image during training. However, these features do not generalize well to a down-stream evaluation task, such as object classification. The desired predictive features of the input image (i.e., the object class and its properties) are suppressed by competing features (i.e., the color distribution of the image). Chen et al. (2021) refer to this phenomenon as *feature suppression among competing features*. Feature suppression among competing features is closely related to work by Hermann & Lampinen (2020), who show that in the presence of multiple redundantly predictive features, deep neural models prefer one of the features over the other, while the other feature is suppressed. Chen et al. (2021) add artificially generated features (i.e., MNIST digits) as an extra overlay to images. They show that the "easy" predictive features (the MNIST digits) are preferred by a deep neural encoder model over the real predictive features (i.e., the object class) when optimizing with a contrastive learning loss. Chen et al. (2021) conclude that contrastive losses rely on easy-to-detect features that solve the contrastive objective, while suppressing the remaining (partly irrelevant) information.

Predictive feature suppression is a prominent problem in resource-constrained contrastive ICR. Captions often describe multiple aspects of a scene. However, in a resource-constrained contrastive setup, only one (or a few) of the aspects that are described in the caption is likely to be sufficient to match with the positive candidate (i.e., the image) during training due to the limited number of negative candidate in the training batch. To mitigate this problem of predictive feature suppression for resource-constrained contrastive ICR, we need an extra optimization objective that is independent of the negative samples in the training batch.

**Autoencoding.** An approach to reduce predictive feature suppression is autoencoding (Hinton & Salakhutdinov, 2006). Autoencoding can be combined with a contrastive learning loss and reduces predictive feature suppression without depending on sampling (hard) negative candidates. To learn high-quality text sequence embeddings for the dense passage retrieval task, Lu et al. (2021) add a weak decoder on top of a document encoder to reconstruct the original document. To make image encoders more robust against shortcut features, Li et al. (2020a) add a decoder on top of the image encoder to decode the input image.

**Upshot**

To reduce predictive feature suppression in a resource-constrained ICR task, we introduce *latent target decoding* (LTD). LTD reduces predictive feature suppression, without focusing on the difficulty of the contrastive discrimination task. LTD requires neither a large number of negative samples nor hard negative mining strategies. Unlike other methods that reconstruct the input data, we reconstruct the input information of the caption in a latent space instead of the input space.

## 3  Method

In Table 4 in Appendix A, we provide an overview of the symbols and variables used throughout this work. We start with preliminaries and then discuss the InfoNCE contrastive loss and why autoencoding captions in the input space is not a solution to reduce predictive feature suppression. Finally, we introduce latent target decoding (LTD) to reduce predictive feature suppression for recourse-constrained ICR. In Figure 2 we provide an overview of LTD.

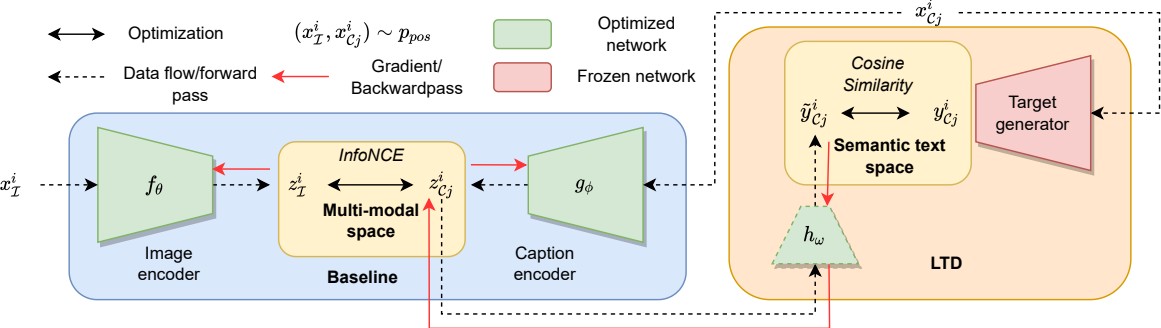

Figure 2: Overview of latent target decoding (LTD). The baseline (left) consists of a general image-caption retrieval framework with an image and caption encoder. The encoders are trained by using the contrastive InfoNCE loss. To reduce predictive feature suppression we add latent target decoding to the baseline ICR method (right). This extra decoder decodes the information of the input caption in a latent space of a general-purpose sentence encoder. The decoder is not used during inference, which we indicate by the dashed line around the model $h_\omega(\cdot)$.

### 3.1  Preliminaries and notation

#### 3.1.1  Notation

We follow the notation introduced in previous work (Chen et al., 2020c; Brown et al., 2020). For the ICR task we use a multi-modal dataset $\mathcal{D} = \{(\boldsymbol{x}_\mathcal{I}^i, \boldsymbol{x}_{\mathcal{C}1}^i, \ldots, \boldsymbol{x}_{\mathcal{C}k}^i), \ldots\}_{i=1}^N$. This dataset consists of $N$ image-caption tuples. Each tuple contains one image $\boldsymbol{x}_\mathcal{I}^i$ and $k$ captions $\boldsymbol{x}_{\mathcal{C}j}^i$, where $1 \leq j \leq k$, that describe the scene depicted in the image. At each training iteration, we randomly sample a batch $\mathcal{B}$ of image-caption pairs from $\mathcal{D}$. Per image, one of the $k$ captions is sampled per training iteration; together, this image and caption form a positive (or matching) image-caption pair. Each caption is used once during a training epoch.

The image and caption encoder are trained for two tasks: *image-to-text* (i2t) and *text-to-image* (t2i) retrieval. Thus, each image and caption in $\mathcal{B}$ is used as a query $\boldsymbol{q}$. We denote the matching candidate in the other modality as $\boldsymbol{v}^+$. All other candidates in $\mathcal{B}$, in the other modality, are considered as negative candidates $\boldsymbol{v}^-$. The set of all negative candidates for query $\boldsymbol{q}$ in batch $\mathcal{B}$ is $\mathcal{S}_{\boldsymbol{q}}^-$, where $\boldsymbol{v}^- \in \mathcal{S}_{\boldsymbol{q}}^-$.

#### 3.1.2  Contrastive baseline model

The baseline (BL) ICR framework in this work consists of two encoders. The image encoder $f_\theta(\cdot)$ takes an image $\boldsymbol{x}_\mathcal{I}^i$ as input and encodes this image into a latent representation $\boldsymbol{z}_\mathcal{I}^i := f_\theta(\boldsymbol{x}_\mathcal{I}^i)$. The caption encoder

$g_\phi(\cdot)$ takes a caption as input and encodes this caption into a latent representation $\boldsymbol{z}_{\mathcal{C}j}^i := g_\phi(\boldsymbol{x}_{\mathcal{C}j}^i)$. The vectors $\boldsymbol{z}_{\mathcal{C}j}^i$ and $\boldsymbol{z}_{\mathcal{I}}^i$ have the same dimensionality and are normalized on the unit sphere. The encoders are only optimized by minimizing a contrastive learning loss $\mathcal{L}_{con}$. Our goal is not to obtain the highest possible evaluation performance, but to show the strength of LTD on resource-constrained training setups.

## 3.2 Contrastive loss

To train the image and caption encoder, we use the InfoNCE contrastive loss (van den Oord et al., 2018; Chen et al., 2020c). The InfoNCE loss is a popular loss function for self-supervised representation learning (Chen et al., 2020c; He et al., 2020) and multi-modal representation learning (Radford et al., 2021; Yuan et al., 2021; Jia et al., 2021). To keep the notation simple and intuitive, we use $\boldsymbol{q}$ and $\boldsymbol{v}$ for the latent representations computed by the caption and image encoder and not $\boldsymbol{z}_{\mathcal{C}j}$ and $\boldsymbol{z}_{\mathcal{I}}$. The InfoNCE loss is defined as follows:

$$\mathcal{L}_{con} = \frac{1}{|\mathcal{B}|} \sum_{(\boldsymbol{q}, \boldsymbol{v}^+) \in \mathcal{B}} - \log \frac{\exp(\boldsymbol{q}^T \boldsymbol{v}^+ / \tau)}{\exp(\boldsymbol{q}^T \boldsymbol{v}^+ / \tau) + \sum_{\boldsymbol{v}^- \in \mathcal{S}_{\boldsymbol{q}}^-} \exp(\boldsymbol{q}^T \boldsymbol{v}^- / \tau)}. \tag{1}$$

$\mathcal{L}_{con}$ in Eq. 1 is minimized when, given a query $\boldsymbol{q}$, the cosine similarity score with the positive candidate $\boldsymbol{v}^+$ is high (i.e., $\approx 1$), while the similarity scores with the negative candidates $\boldsymbol{v}^- \in \mathcal{S}_{\boldsymbol{q}}^-$ in the batch are as low as possible; $\tau$ serves as a temperature parameter. The main objective of a contrastive learning loss for the ICR task is to learn data representations that can be used to discriminate between similar and dissimilar image-caption pairs. However, there is no constraint that enforces the encoders to learn representations that contain all available input information to make this discrimination, which is what we add.

### 3.2.1 Gradient w.r.t. the input representations

To show some important properties of the InfoNCE loss, we provide the derivative of $-\mathcal{L}_{con}$ w.r.t. the input in Eq. 2 (Chen et al., 2020c):

$$Z(\boldsymbol{q}, \boldsymbol{v}) = \frac{\exp(\boldsymbol{q}^T \boldsymbol{v} / \tau)}{\exp(\boldsymbol{q}^T \boldsymbol{v}^+ / \tau) + \sum_{\boldsymbol{v}^- \in \mathcal{S}_{\boldsymbol{q}}^-} \exp(\boldsymbol{q}^T \boldsymbol{v}^- / \tau)} \tag{2a}$$

$$\frac{\partial \mathcal{L}_{con}}{\partial \boldsymbol{q}} \tau = (1 - Z(\boldsymbol{q}, \boldsymbol{v}^+)) \boldsymbol{v}^+ - \sum_{\boldsymbol{v}^- \in \mathcal{S}_{\boldsymbol{q}}^-} Z(\boldsymbol{q}, \boldsymbol{v}^-) \boldsymbol{v}^- \tag{2b}$$

$$\frac{\partial \mathcal{L}_{con}}{\partial \boldsymbol{v}^+} \tau = (1 - Z(\boldsymbol{q}, \boldsymbol{v}^+)) \boldsymbol{q} \tag{2c}$$

$$\frac{\partial \mathcal{L}_{con}}{\partial \boldsymbol{v}^-} \tau = -Z(\boldsymbol{q}, \boldsymbol{v}^-) \boldsymbol{q}. \tag{2d}$$

$Z(\boldsymbol{q}, \boldsymbol{v})$ returns the similarity score of candidate $\boldsymbol{v}$ w.r.t. the query $\boldsymbol{q}$, normalized by the sum of similarity scores of all candidates in the batch $\mathcal{B}$. The full derivations of the derivative $-\mathcal{L}_{con}$ are provided in Appendix B. Based on Eq. 2, we infer the following properties:

1. The update w.r.t. the query $\boldsymbol{q}$ (Eq. 2b), is a weighted sum over the positive candidate $\boldsymbol{v}^+$ and all negatives $\boldsymbol{v}^- \in \mathcal{S}_{\boldsymbol{q}}^-$. The query representation $\boldsymbol{q}$ will be pulled closer to $\boldsymbol{v}^+$, while being pushed away from all $\boldsymbol{v}^- \in \mathcal{S}_{\boldsymbol{q}}^-$. The weight value of each candidate, $Z(\boldsymbol{q}, \boldsymbol{v})$ (Eq. 2a), depends on the similarity score with the query.
2. $\boldsymbol{v}^+$ (Eq. 2c) will be pulled closer to the query representation (and the other way around).
3. All negatives $\boldsymbol{v}^-$ (Eq. 2d) will be pushed away from the query representation (and the other way around).

Without contrasting with negative candidates, the encoders will learn a trivial solution where latent representations collapse to a single point in the latent space (Jing et al., 2021). This means that the learned representation mainly depends on contrasting with negative candidates during training. If the representations $\boldsymbol{v}$ only contain a subset of the predictive input features (which still minimize the contrastive training objective), the query representation $\boldsymbol{q}$ (in the other modality) will be updated to match/mismatch these representations.

The contrastive InfoNCE objective itself does not guarantee that all the predictive features in the input data are learned (Robinson et al., 2021) and mainly relies on easy-to-detect features to contrast between positive and negative pairs (Chen et al., 2021).

Importantly, the query and candidate representations are in different modalities and therefore generated by different encoders. Hence, the update of the query and candidate representations is based on *fixed* representations in the other modality (e.g., the caption encoder can only try to match/not match with the representations of the image encoder and vice versa). By adding a constraint on the representations of one of the two modalities, the other modality encoder will follow automatically. Therefore, in order to prevent predictive feature suppression for the caption modality in a resource-constraint ICR setting, we add a constraint to the learning algorithm that forces the caption representation to be projected into the latent space of a general-purpose sentence encoder.

### 3.3 Autoencoding reconstruction objective

Autoencoding (Hinton & Salakhutdinov, 2006) is a natural choice for learning latent data representations that contain most of the important input features without relying on hard negative samples. To reconstruct the input caption from the encoded latent representation $\boldsymbol{z}_{\mathcal{C}j}^{i}$, we introduce a decoder network $h_{\omega}(\cdot)$:

$$\widetilde{\boldsymbol{x}}_{\mathcal{C}j}^{i} \coloneqq h_{\omega}(\boldsymbol{z}_{\mathcal{C}j}^{i}). \tag{3}$$

The decoder network $h_{\omega}(\cdot)$ takes the latent caption representation as input and outputs a reconstruction of the input caption $\widetilde{\boldsymbol{x}}_{\mathcal{C}j}^{i}$. To decode the input sequence from the latent representation, this latent representation should be a dense representation of the entire input sequence. The reconstruction loss, $\mathcal{L}_{rec}$, of a sequence of tokens, $x_i, \ldots, x_n$ of length $n$, is the negative log-likelihood of the input data:

$$\mathcal{L}_{rec} = -\sum_{t=1}^{n} \log\ p(x_t|x_{t-1:1}, \boldsymbol{z}_{\mathcal{C}j}^{i}). \tag{4}$$

Based on Eq. 4 it is clear that each predicted token $x_t$ in the sequence is conditioned on: (i) the latent caption representation $\boldsymbol{z}_{\mathcal{C}j}^{i}$, and (ii) the already predicted sequence $x_{t-1:1}$.

As discussed in Section 1, a strong decoder will mainly rely on the learned language model and language patterns to decode the input sequence (Lu et al., 2021). This implies that the input sequence can be decoded correctly while mainly ignoring $\boldsymbol{z}_{\mathcal{C}j}^{i}$, especially when $t$ is large. Therefore, decoding the caption sequence in the input space is not guaranteed to reduce predictive feature suppression.

### 3.4 Latent target decoding

In Section 3.2 we argued why the contrastive InfoNCE loss is prone to predictive feature suppression, and in Section 3.3 we discussed why decoding a caption in the input space will not prevent predictive feature suppression. In this section, we introduce *latent target decoding* (LTD). LTD decodes the semantics of the input caption in a latent space of a general-purpose sentence encoder to reduce predictive feature suppression, which can be used in combination with a contrastive loss. LTD addresses the issues of decoding the caption in the input space. See Figure 2 for a high-level overview of LTD for ICR.

For each caption $\boldsymbol{x}_{\mathcal{C}j}^{i}$ in the training dataset we generate $\boldsymbol{y}_{\mathcal{C}j}^{i}$, a *latent target representation*. The vector $\boldsymbol{y}_{\mathcal{C}j}^{i}$ is a dense vector representation. We assume that this vector contains all the (semantic) information of the caption, captured by a general-purpose language encoder. We use our decoding network $h_{\omega}$ to decode $\boldsymbol{y}_{\mathcal{C}j}^{i}$ instead of the input caption. By reconstructing a vector representation of the caption instead of the original input sequence, the reconstruction is not conditioned on the already predicted sequence of tokens. The latent target decoder assumes conditional independence of each feature in the latent target. Therefore, the decoder cannot rely on conditional (language model) patterns in the data to reconstruct the input semantics. This implies that we force the decoder to rely completely on $\boldsymbol{z}_{\mathcal{C}j}^{i}$ to decode the latent target representation. LTD reduces predictive feature suppression by reconstructing the latent target from the caption embedding. To combine LTD with a contrastive loss, it is necessary to compute the similarity score between a *global*

representation of the entire caption and the image embedding. ICR methods that compute the similarity score by using fragments of the caption and regions in the image cannot be combined with LTD as introduced in this work. If there is no global representation of the caption, it is not possible enforce that all the semantic information from the target encoder will be distilled into the caption representation that is used for computing the similarity score.

### 3.4.1  Target decoding network

To decode $\boldsymbol{y}^i_{\mathcal{C}j}$ we use a three layer feed-forward decoder network:

$$h_\omega(\boldsymbol{z}^i_{\mathcal{C}j}) = \boldsymbol{W}^{(3)}\sigma\left(\boldsymbol{W}^{(2)}\sigma\left(\boldsymbol{W}^{(1)}\ \boldsymbol{z}^i_{\mathcal{C}j}\right)\right), \tag{5}$$

where $\sigma$ is the ReLU non-linearity; $h_\omega$ takes the latent caption representation $\boldsymbol{z}^i_{\mathcal{C}j}$ as input and maps it to a reconstruction of the latent target representation $\widetilde{\boldsymbol{y}}^i_{\mathcal{C}j}$.

### 3.4.2  Loss function

To train $h_\omega$, we use the cosine distance between $\widetilde{\boldsymbol{y}}^i_{\mathcal{C}j}$ and $\boldsymbol{y}^i_{\mathcal{C}j}$ as *reconstruction loss* $\mathcal{L}_{rec}$:

$$\mathcal{L}_{rec} = 1 - \frac{\widetilde{\boldsymbol{y}}^i_{\mathcal{C}j}}{\left\|\widetilde{\boldsymbol{y}}^i_{\mathcal{C}j}\right\|} \cdot \frac{\boldsymbol{y}^i_{\mathcal{C}j}}{\left\|\boldsymbol{y}^i_{\mathcal{C}j}\right\|}. \tag{6}$$

Minimizing the cosine distance is equivalent to minimizing the mean squared error of two vectors normalized on the unit sphere (Chen & He, 2021; Grill et al., 2020). By introducing an extra loss criterion, the overall training objective becomes a dual optimization problem. The *dual loss* $\mathcal{L}_{dual}$ is defined as follows:

$$\mathcal{L}_{dual} = \mathcal{L}_{con} + \beta\mathcal{L}_{rec}, \tag{7}$$

where $\beta$ serves as a balancing parameter to scale the two losses.

### 3.4.3  Constraint-based optimization

By adding LTD to the learning framework, we introduce an extra loss component $\mathcal{L}_{rec}$. To effectively minimize $\mathcal{L}_{dual}$, we have to find the right value for the balancing parameter $\beta$ in Eq. 7. This may require a considerable amount of manual tuning, and often one specific value for $\beta$ does not generalize to different training settings. Besides that, $\mathcal{L}_{rec}$ is not the main training objective for the ICR tasks. The main reason we add $\mathcal{L}_{rec}$ to the learning algorithm is to reduce predictive feature suppression caused by solely optimizing the contrastive loss. We therefore argue that implementing LTD as an optimization *constraint* (Rezende & Viola, 2018; van Rozendaal et al., 2020), as opposed to an optimization *objective*, might be more effective. Our goal, then, is to minimize the contrastive loss $\mathcal{L}_{con}$ given the constraint that the reconstruction loss is lower than or equal to a certain bound value $\eta$:

$$\min_{\theta,\psi,\omega} \mathcal{L}_{con} \text{ subject to } \mathcal{L}_{rec} \leq \eta. \tag{8}$$

We can implement this optimization constraint in combination with gradient descent by using the method of Lagrange multipliers:

$$\max_\lambda \min_{\theta,\psi,\omega} \mathcal{L}_{lag} = \mathcal{L}_{con} + \lambda\left(\frac{\mathcal{L}_{rec}}{\eta} - 1\right). \tag{9}$$

The optimization objective is to minimize $\mathcal{L}_{lag}$ w.r.t. the model parameters $\theta, \psi, \omega$, while maximizing $\mathcal{L}_{rec}$ w.r.t. to the multiplier $\lambda$. The multiplier $\lambda$ is tuned automatically by using stochastic gradient ascent with momentum. By optimizing $\lambda$ with stochastic gradient ascent, the two losses will be balanced automatically during training such that the reconstruction constraint is met, while the contrastive loss is minimized by gradient descent.

### 3.4.4 Choice of latent target representation

To generate the latent target $\boldsymbol{y}_{\mathcal{C}j}$, we use a Sentence-BERT transformer model (Reimers & Gurevych, 2019; Song et al., 2020).[1] Sentence-BERT is a general-purpose sentence encoder that is trained on a large amount of data to capture the semantic input information. Thus, we expect these embeddings to be more general than those we learn for the resource-constrained contrastive ICR task, which makes them a suitable choice for the latent target representations $\boldsymbol{y}_{\mathcal{C}j}$.

### 3.5 LTD vs. teacher-student framework

LTD is somewhat similar to a teacher-student framework used with knowledge distillation (Hinton et al., 2015). Indeed, the target generator can be seen as a teacher network and the caption encoder in combination with the target decoder as a student network. However, in contrast with knowledge distillation, the goal of LTD is *not* to closely mimic a teacher network. Instead, the goal is to learn caption representations that can be used for multi-modal contrastive-based retrieval while extracting as much of the textual semantic input information of the caption as possible.

## 4 Experimental setup

We design experiments aimed at showing: (i) a reduction of predictive feature suppression by using LTD, with a focus on the ICR task; (ii) the advantages of LTD over reconstructing the caption in the input space; (iii) the benefit of constraint-based optimization of LTD over dual loss optimization; and (iv) the generalizability of LTD to different contrastive losses and resource-constrained ICR methods that use different encoder network architectures. To facilitate reproducibility and further research of our work, we include the code with our paper.[2]

### 4.1 Datasets

For training and evaluating our ICR method, we use the two common ICR benchmark datasets: Flickr30k (F30k) (Young et al., 2014) and MS-COCO captions (COCO) (Lin et al., 2014). The F30k dataset contains 31,000 image-caption tuples. We use the train, validate and test split from (Karpathy & Fei-Fei, 2015), with 29,000 images for training, 1,000 for validation, and 1,000 for testing. COCO consists of 123,287 image-caption tuples. We use the train, validate and test split from (Karpathy & Fei-Fei, 2015); we do not use the 1k test setup. Both F30k and COCO come with $k = 5$ captions per image.

We also use the crisscrossed captions (CxC) dataset, which extends the COCO validation and test set with additional annotations of similar captions and images (Parekh et al., 2020), so as to evaluate whether LTD improves the evaluation scores by retrieving semantically similar candidates.

### 4.2 Implementation details

Unless otherwise specified, we use the following architectures for the target decoder, image encoder, and caption encoder. We use similar network architectures for the image and caption encoder as the ones used in (Chun et al., 2021), which are simple network architectures that can be trained using a limited amount of training data.

**Image encoder.** For the image encoder, we use a pre-trained ResNet-50 (He et al., 2016) network. We apply average pooling on the last convolutional layer followed by a projection head to map the image feature vector into a shared multi-modal latent space; the projection head has two feed-forward layers and a ReLU non-linearity.

**Caption encoder.** For the caption encoder, we use a bi-directional, single-layer, GRU network (Cho et al., 2014). We use pre-trained GloVe embeddings (Pennington et al., 2014) as word embeddings. We use a similar

---

[1]https://huggingface.co/sentence-transformers/all-mpnet-base-v2
[2]https://github.com/MauritsBleeker/reducing-predictive-feature-suppression/

projection head as for the image encoder (which does not share parameters) to map the caption embedding into the shared latent space.

**Target decoding network.** For the target decoding network, we use a three-layer feed-forward network as in Eq. 5. To generate the latent target representations, we use the HuggingFace *all-MiniLM-L6-v2* Sentence-BERT implementation. The target decoding network is trained together with the image and caption encoder. The target decoding network is *not* used during evaluation.

**Input decoding network.** We compare *latent* target decoding (LTD) with *input* target decoding (ITD), which reconstructs the input caption in the input space (i.e., the input tokens). For ITD, we use a single-layer GRU (Cho et al., 2014) decoder that reconstructs the input tokens in the caption (as explained in Section 3.3). We train the word embeddings for ITD from scratch. ITD is optimized with the negative log-likelihood loss (Eq. 4).

**Training.** Similar to (Chun et al., 2021), we use 30 warm-up and 30 fine-tune epochs, a batch size of 128, and a cosine annealing learning rate schedule with an initial learning rate of $2e^{-4}$. The Lagrange multiplier is initialized with a value of 1, bounded between 0 and 100, and is optimized by stochastic gradient ascent with a fixed learning rate of $5e^{-3}$ and a momentum (to prevent $\lambda$ from fluctuating too much) and dampening value of $\alpha = 0.9$. When we use $\mathcal{L}_{dual}$, we set $\beta$ to 1. For the InfoNCE loss, we use a temperature value $\tau$ of 0.05. Evaluation scores of ICR methods tend to differ depending on the random seed used during training (Rao et al., 2022); to improve robustness, we apply stochastic weight averaging (SWA) (Izmailov et al., 2018); we take the average of 5 checkpoints, stored during the last 10% of the training iterations each epoch. For the reconstruction constraint bound $\eta$, we try for all experiments several values, $\eta \in \{0.05, 0.1, 0.15, 0.2, 0.25, 0.3\}$. When we apply ITD we use $\eta \in \{0.5, 1, 2, 3, 4, 5, 6\}$. All results are based on the best-performing value of $\eta$.

**Generalizibility to different network architectures.** LTD is a general method that can be combined with any global representation contrastive ICR method. To show that LTD works in combination with different network architectures, we apply LTD with multiple ICR methods that can be trained in a resource-constrained setup. To cover a wide spectrum of network architectures, we choose two methods that use different network architectures for the image and caption encoder.

*VSRN.* The visual semantic reasoning network (VSRN) (Li et al., 2019) consists of an image and caption encoder that both compute a global representation of each input modality. The caption encoder consists of a single directed GRU, similar to the caption encoder used in (Faghri et al., 2018). The image encoder takes a set of pre-computed regions of interest as input generated by a ResNet-101 (He et al., 2016) backbone, pre-trained on visual genome (Krishna et al., 2017). This set of pre-computed visual features is considered a fully connected graph of regions in the input image. To perform reasoning on the graph of visual features, a multi-layer graph convolutional networks (Kipf & Welling, 2017) is used. Finally, to obtain one global representation of the entire image, a GRU is used to aggregate the regions of interest into one single representation. We use the same learning rate schedule and number of training epochs as in (Li et al., 2019) and we use the model implementation as provided by the authors.[3] However, we modify the original VSRN model on two points:

(i) We use the same caption encoder as described in Section 4.2 instead of the *single* directed GRU used for the original VSRN model and use a hidden dimensional of 1024 instead of 2048 for the caption and image representations. The goal of this work is not to show which specific network architectures perform best for the ICR task, but to show the generalizability of LTD in combination with different encoder networks.

(ii) The original VSRN model also comes with an additional caption decoder that decodes the input caption from the visual features. In this work, we investigate the reduction of predictive feature suppression for the general ICR framework, consisting of two encoders optimized by using a contrastive loss. If we would add LTD to the original VSRN method, we would have two reconstruction objectives and a contrastive loss. The main reason we use VSRN is for the use of graph convolution networks

---

[3] https://github.com/KunpengLi1994/VSRN

in the image encoder to show the generalizability of LTD in combination with different encoder networks. Therefore, we remove this caption decoder from the learning algorithm.

*TERN.* The transformer reasoning network (TERN) (Messina et al., 2020b) is a transformer-based ICR method that is solely trained and evaluated on the COCO dataset. TERN consists of a pre-trained BERT (Devlin et al., 2018) caption encoder. The image encoder takes a set of pre-computed regions of interest as input (similar to VSRN) and consists of a stack of four transformer layers. The pre-computed features are only available for the COCO dataset. Next, both the image and caption features are pushed through a stack of shared transformer layers. Although the weights of the last part of the caption and image encoder are shared, there is no (cross) attention between the two modalities; the representations of the images and captions are still computed independently. The image and caption CLS token is used as a global representation of both the image and the caption. We use the same learning rate schedule, dropout rate, number of training epochs, and data augmentations as in (Messina et al., 2020b) and use the model implementation as provided by the authors.[4]

In this work, we use ICR methods that are trained from scratch on the F30k and COCO dataset and that are trainable on a single GPU. That does not imply that some of the weights of our encoders are not initialized with pre-trained parameters. However, we only use pre-trained weights that are trained on a uni-modal task(s), and not for image-text matching specifically.

## 4.3 Evaluation metrics

To measure the reduction of predictive feature suppression, we evaluate how well the learned encoders generalize to the ICR evaluation task. The more predictive features the encoders learn to capture, the better these encoders are able to retrieve the correct candidate given a query. The standard evaluation metric for ICR is recall@$k$ metric, with $k = \{1, 5, 10\}$. During training, we evaluate the model after each training epoch on the validation set. Similar to (Faghri et al., 2018; Lee et al., 2018; Li et al., 2019; Chen et al., 2020a; Chun et al., 2021), we select the model with the highest score on the validation set (using the sum of all recall@$k$ scores as a metric) for evaluation on the test set.

**Recall@$k$.** For ICR, recall@$k$ is implemented as the fraction of how many times a matching candidate is present in the top-$k$ ranking (Karpathy & Fei-Fei, 2015; Parekh et al., 2020). For the t2i task, there is only one matching image per query caption. For the i2t task, there are 5 matching captions per image. Only the highest-ranked caption per query is used for calculating the recall scores. When using the CxC annotations, for both i2t and t2i, we take the highest-ranked candidate.

**R-precision.** When extending the COCO dataset with the CxC annotations, we have one or more matching candidates per query for both i2t and t2i. Like Chun et al. (2021), we also use r-precision (R-P) for evaluation; it measures the precision for a top-$r$ ranking, where $r$ is the number of matching candidates given a query.

**nDCG.** The standard evaluation metric for the ICR task, recall@$k$, mainly measures if the positive candidate is present in the top $k$ of the ranking. However, this does not provide much insight into the overall quality of the ranking. To address the limitations of only using recall@$k$, Messina et al. (2020b) start using nDCG as an additional evaluation metric for t2i retrieval. However, for t2i retrieval there is only one positive image per query caption. To generate more positive images per caption query, images that have captions with a high overlap with the query caption, are also considered positive. As similarity measurements between the captions, ROUGE-L (Lin, 2004) and SPICE (Anderson et al., 2016) are used. There are multiple re-annotations of COCO available that provide multiple matching images per caption query; see, for example, (Parekh et al., 2020). However, these re-annotations are not used by Messina et al. (2020b) to compute the nDCG scores. To keep the evaluation consistent, we use the same relevance labels as used in (Messina et al., 2020b). The nDCG relevance labels are only available for COCO and not for F30k.

---

[4]`https://github.com/mesnico/TERN`

| | | | i2t | | | | t2i | | | | nDCG | |
|---|---|---|---|---|---|---|---|---|---|---|---|---|
| | | | R@k | | | | R@k | | | | | |
| # | Method | Loss | 1 | 5 | 10 | R-P | 1 | 5 | 10 | R-P | ROUGE-L | SPICE |
| | | | | | | F30k | | | | | | |
| 1.1 | BL | $\mathcal{L}_{con}$ | 47.4 | 75.9 | 84.8 | 0.34 | 33.9 | 65.2 | 76.6 | - | - | - |
| 1.2 | BL + ITD | $\mathcal{L}_{dual}, \beta=1$ | 45.7 | 74.0 | 84.4 | 0.33 | 33.7 | 65.1 | 75.8 | - | - | - |
| 1.3 | BL + ITD | $\mathcal{L}_{lag}, \eta=6$ | 36.6 | 66.8 | 76.5 | 0.28 | 27.8 | 59.1 | 71.0 | - | - | - |
| 1.4 | BL + LTD | $\mathcal{L}_{dual}, \beta=1$ | 46.1 | 75.3 | 84.1 | 0.34 | 34.0 | 65.9 | 77.4 | - | - | - |
| 1.5 | BL + LTD | $\mathcal{L}_{lag}, \eta=0.2$ | **49.6** | **78.7** | **86.4** | **0.37** | **36.7** | **68.4** | **79.3** | - | - | - |
| | | | | | | COCO | | | | | | |
| 2.1.1 | BL | $\mathcal{L}_{con}$ | 33.7 | 64.4 | 76.6 | 0.24 | 24.2 | 53.5 | 67.0 | - | 0.6487 | 0.5729 |
| 2.2.1 | BL + ITD | $\mathcal{L}_{dual}, \beta=1$ | 32.7 | 64.4 | 76.3 | 0.24 | 24.2 | 53.8 | 67.6 | - | 0.6496 | 0.5733 |
| 2.3.1 | BL + ITD | $\mathcal{L}_{lag}, \eta=4$ | 28.4 | 59.2 | 72.2 | 0.22 | 22.0 | 50.4 | 64.5 | - | 0.6424 | 0.5638 |
| 2.4.1 | BL + LTD | $\mathcal{L}_{dual}, \beta=1$ | 34.2 | 64.7 | 76.6 | 0.25 | 25.0 | 54.3 | 67.9 | - | 0.6510 | 0.5756 |
| 2.5.1 | BL + LTD | $\mathcal{L}_{lag}, \eta=0.15$ | **36.0** | **66.5** | **78.1** | **0.26** | **26.2** | **56.2** | **69.4** | - | **0.6531** | **0.5786** |
| | | | | | | CxC | | | | | | |
| 2.1.2 | BL | $\mathcal{L}_{con}$ | 36.1 | 68.1 | 80.2 | 0.22 | 26.7 | 57.6 | 71.0 | 0.23 | - | - |
| 2.2.2 | BL + ITD | $\mathcal{L}_{dual}, \beta=1$ | 35.0 | 68.0 | 79.7 | 0.22 | 26.6 | 58.0 | 71.6 | 0.23 | - | - |
| 2.3.2 | BL + ITD | $\mathcal{L}_{lag}, \eta=4$ | 31.0 | 62.9 | 75.8 | 0.20 | 24.6 | 54.8 | 68.9 | 0.21 | - | - |
| 2.4.2 | BL + LTD | $\mathcal{L}_{dual}, \beta=1$ | 36.6 | 68.1 | 79.9 | 0.23 | 27.6 | 58.8, | 72.0 | 0.24 | - | - |
| 2.5.2 | BL + LTD | $\mathcal{L}_{lag}, \eta=0.15$ | **38.4** | **70.4** | **81.5** | **0.24** | **28.9** | **60.4** | **73.3** | **0.25** | - | - |

Table 1: Recall@k, nDCG, and r-precision (R-P) evaluation scores for the F30k and COCO datasets (including the CxC annotations). We evaluate three loss functions $\mathcal{L}_{con}$ , $\mathcal{L}_{dual}$ and $\mathcal{L}_{lag}$. We use three methods, the contrastive ICR baseline (BL), BL + input target decoding (ITD), and BL + latent target decoding (LTD). Boldface indicates the highest value for each evaluation metric per dataset. '-' indicates that it is not possible to compute the evaluation score for that dataset/experiment since the annotations are not available.

# 5 Results

In Section 5.1, we compare a contrastive ICR baseline with the same baseline combined with LTD. Next, in Section 5.2 we ask if similar results can be achieved with ITD as with LTD. In Section 5.3, we investigate the role of the optimization constraint and compare constraint-based LTD with LTD optimized as a dual loss. Finally, in Section 5.4, we ask whether LTD can be used in combination with a different contrastive loss function, and in Section 5.5 we show that LTD can be combined with a wide variety of resource-constrained ICR methods.

## 5.1 Constrastive ICR baseline vs. baseline + LTD

In Table 1 we compare the contrastive ICR baseline, which is optimized by using the contrastive loss $\mathcal{L}_{con}$ defined in Eq. 1, with the baseline combined with LTD. Based on Table 1, row 1.1, 1.4, 2.1.1, 2.4.1, and 2.4.2, 2.4.2, we observe that LTD optimized as a dual loss ($\mathcal{L}_{dual}$) with $\beta = 1$ does not convincingly (or only with a small margin) outperform the baseline ICR method, which is optimized solely in a contrastive manner, in terms of recall@$k$, nDCCG, and r-precision for both datasets and both i2t and t2i.

In contrast, when we implement LTD as an optimization constraint, by using $\mathcal{L}_{lag}$, row 1.5, 2.1.5, and 2.2.5, we observe that LTD consistently outperforms the baseline ICR methods on both F30k and COCO for both tasks (i2t and t2i) with a large margin. An increase in recall also comes with an increase in the r-precision

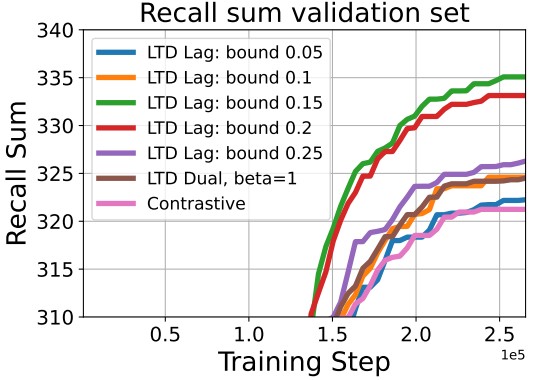

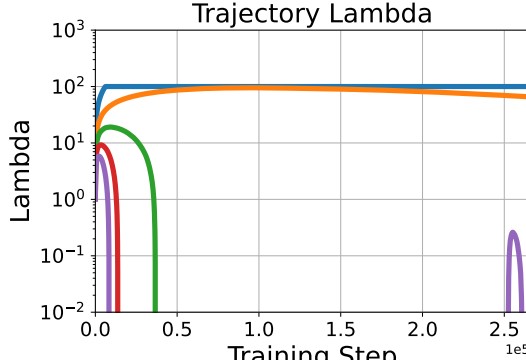

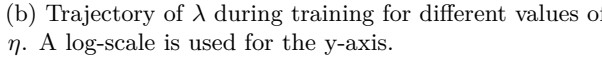

(a) Trajectory of the recall sum on the validation set during training.

(b) Trajectory of $\lambda$ during training for different values of $\eta$. A log-scale is used for the y-axis.

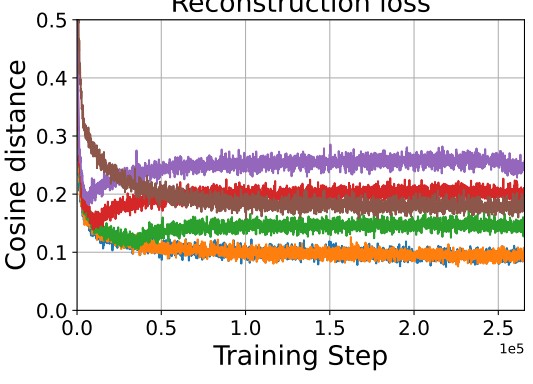

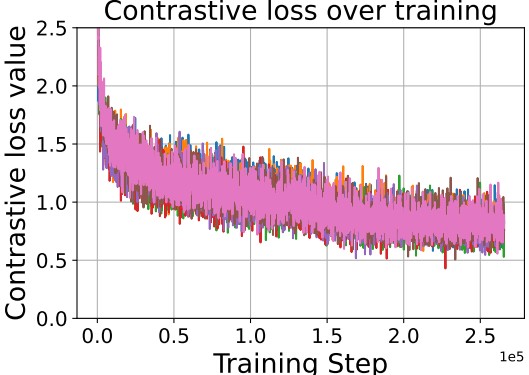

(c) Trajectory of the reconstruction loss $\mathcal{L}_{rec}$.

(d) Trajectory of the contrastive loss $\mathcal{L}_{con}$.

Figure 3: Overview of constraint-based optimization on the evaluation metric and the optimization objectives. We train $\mathcal{L}_{lag}$ with four different values of $\eta \in \{0.05, 0.10, 0.15, 0.20, 0.25\}$. All training steps are to the power of $1e5$.

scores and nDCG scores. Hence, features learned by constraint-based LTD perform better on the evaluation task, which is an indication of the reduction of predictive feature suppression.

## 5.2 Latent Target Decoding vs. Input Target Decoding

As argued in Section 3.3, decoding the caption in the input space will probably not result in a reduction of predictive feature suppression due to overfitting of the learned language model. To empirically show this, we also implemented a decoder that decodes tokens of the input caption (ITD) to reduce predictive feature suppression (see Section 4.2 for details). Based on row 1.2, 2.1.2, and 2.2.2 in Table 1, we conclude that implementing ITD as a dual loss does not result in improved recall@$k$ scores, for most values of $k$, compared to the contrastive baseline. Surprisingly, when we implement ITD as an optimization constraint (with $\eta = 4$ for F30k and $\eta = 6$ for COCO, other values of $\eta$ do not yield improvements) the evaluation scores are even lower (row 1.3, 2.1.3 and 2.2.3) than when implemented as a dual loss. We conclude that: (i) ITD does not reduce predictive feature suppression for ICR, and (ii) implementing ITD as an optimization constraint even hurts performance.

## 5.3 The role of the optimization constraint

What is the role of the optimization constraint when minimizing $\mathcal{L}_{rec}$ and what is the effect on the evaluation scores compared to using $\mathcal{L}_{dual}$? In Figure 3b we plot the trajectory of $\lambda$ for different values of $\eta \in$

$\{0.05, 0.1, 0.15, 0.2, 0.25\}$ during training on the COCO dataset. We also provide (i) the trajectory of the evaluation score (recall sum) over the validation set during training (Figure 3a), (ii) the trajectory of the reconstruction loss for different values of $\eta$ and when optimized without using a constraint ($\mathcal{L}_{dual}$) (Figure 3c), and (iii) the trajectory of the contrastive loss for different values of $\eta$ (Figure 3d). Based on Figure 3 we observe:

1. $\lambda$ increases until the optimization constraint is met (i.e., the bound $\eta$). The closer the reconstruction loss is to $\eta$, the slower the increase of $\lambda$. When the reconstruction constraint is met, $\lambda$ decreases to 0 (Figure 3b).
2. $\lambda$ is positive again when the reconstruction loss becomes higher than the bound $\eta$ (Figure 3b, purple line).
3. The reconstruction loss converges to the value of $\eta$ (Figure 3c). However, it is not possible to meet every value of $\eta$. E.g., $\eta = 0.05$ is too low to achieve for the model.
4. A lower reconstruction loss does not necessarily result in higher evaluation scores (Figure 3a). E.g., the recall sum is higher for $\eta = 0.15$ than for $\eta = 0.1$ or $\eta = 0.05$.
5. The value and the development of the contrastive loss do not depend much on the value of the reconstruction loss (Figure 3d). E.g., a model optimized with $\mathcal{L}_{con}$ has the same contrastive loss trajectory as a model that is optimized with $\mathcal{L}_{lag}$ and $\mathcal{L}_{dual}$. Hence, the contrastive loss on its own does not provide a good indication of the performance on the evaluation task. Similar trajectories of the contrastive loss result in different evaluation scores (hence different learned representations).

When we implement LTD as a dual loss, there is always a gradient from the reconstruction loss w.r.t. the parameters of the caption encoder, until the reconstruction loss is 0. This is not the case when we implement LTD as a reconstruction constraint. When the constraint is met, $\lambda$ drops to zero and there is no gradient anymore from the reconstruction loss. We can conclude that a constant gradient from the reconstruction loss does not improve the learned representations of the caption encoder in terms of evaluation scores. The evaluation scores are higher when there is only a gradient until a certain reconstruction bound $\eta$ is met.

### 5.4   Generalizability w.r.t. contrastive loss

In Section 3.2 we argued that the InfoNCE loss is prone to predictive feature suppression. A popular choice of contrastive loss function for ICR methods is the triplet loss with in-batch hard-negative mining (Faghri et al., 2018). The triplet loss with in-batch hard-negative mining is a special case of the InfoNCE loss, where the number of positives and negatives are each one (Khosla et al., 2020). Therefore, our line of reasoning in Section 3.2 holds for the triplet loss too.

To show the strength and generalizability of LTD to other contrastive losses, we run the same experiments as in Section 5.1 (only for LTD not for ITD), with the triplet loss instead of the InfoNCE loss as $\mathcal{L}_{con}$. To prevent the triplet loss from collapsing to the trivial solution, we added a batch normalization layer after the projection head, for both the image and caption encoder; we use a margin value of $\alpha = 0.2$ (Faghri et al., 2018; Li et al., 2019; Messina et al., 2020b). Based on Table 1, we can observe that the highest recall@$k$ scores also come with the highest r-precision and nDCG scores. Since the main goal of this experiment is to show that LTD can be used in combination with different contrastive losses we, therefore, only evaluate for recall@$k$.

Table 2 provides the recall@$k$ scores for the F30k and COCO datasets. For both F30k and COCO the triplet loss with constraint-based LTD (see rows 3.1.3 and 3.2.3) results in higher evaluation scores than the InfoNCE loss with constraint-based LTD (see Table 1, rows 1.5 and 2.5). Our goal here is not to identify the best contrastive loss for ICR or LTD, but to show the generalizability of LTD to different contrastive losses. Moreover, using the triplet loss as $\mathcal{L}_{con}$ (see row 3.2.1), results in expected evaluation scores on the COCO dataset (given the reproducibility work in (Bleeker & de Rijke, 2022)). Surprisingly, however, the evaluation scores for the F30k dataset while using the triplet loss as $\mathcal{L}_{con}$ (see row 3.1.1) are lower than expected (when compared to Table 1, row 1.1). It is unclear why we observe these low evaluation scores for the F30k dataset when only using the triplet loss as $\mathcal{L}_{con}$. In contrast, we observe that constraint-based LTD in combination with the triplet loss drastically improves the evaluation scores for the F30k dataset, which shows the strength

of constraint-based LTD for improving ICR evaluation scores and also making the triplet loss more robust to predictive feature suppression and feature collapsing.

| # | Method | Loss | i2t | | | t2i | | |
|---|--------|------|-----|-----|------|-----|-----|------|
| | | | R@1 | R@5 | R@10 | R@1 | R@5 | R@10 |
| | | | F30k | | | | | |
| 3.1.1 | BL | $\mathcal{L}_{con}$ | 12.8 | 33.2 | 45.4 | 11.1 | 32.6 | 46.6 |
| 3.1.2 | BL + LTD | $\mathcal{L}_{dual}, \beta = 1$ | 17.1 | 42.7 | 56.4 | 13.1 | 40.5 | 55.7 |
| 3.1.3 | BL + LTD | $\mathcal{L}_{lag}, \eta = 0.2$ | **54.7** | **81.5** | **88.3** | **40.8** | **71.3** | **80.6** |
| | | | COCO | | | | | |
| 3.2.1 | BL | $\mathcal{L}_{con}$ | 37.1 | 67.1 | 78.2 | 27.8 | 56.6 | 69.3 |
| 3.2.2 | BL + LTD | $\mathcal{L}_{dual}, \beta = 1$ | 37.4 | 67.8 | 79.1 | 28.2 | 57.2 | 70.5 |
| 3.2.3 | BL + LTD | $\mathcal{L}_{lag}, \eta = 0.2$ | **39.1** | **69.3** | **80.6** | **29.6** | **59.4** | **72.2** |

Table 2: Recall@$k$ evaluation scores for the F30k and COCO datasets. For experiments 3.*, we use the triplet loss with in-batch hard-negative mining, as defined in (Faghri et al., 2018) instead of the InfoNCE loss (van den Oord et al., 2018) for $\mathcal{L}_{con}$. Boldface indicates the highest value for each evaluation metric per dataset.

## 5.5 Generalizability w.r.t. network architectures

In this section, we consider whether LTD can be used in combination with different resource-constrained ICR methods that use different network architectures. To answer this question we use LTD in combination with the VSRN and TERN methods.

In line with the observations in Table 1, we observe in Table 3 that for both VSRN and TERN: (i) constraint-based LTD outperforms the contrastive baseline, and (ii) constraint-based LTD results in a stronger performance improvement than implementing LTD as a dual loss.

In line with the results in (Messina et al., 2020b), the VSRN baseline outperforms the TERN baseline in terms of Recall@$k$. However, the difference in nDCG scores between the two models is relatively small. Although constraint-based LTD outperforms both the baseline and LTD implemented as a dual loss, improvements gained by using LTD in combination with VSRN are less convincing than for TERN.

The most consistent improvement in evaluation scores is obtained by combining LTD with TERN, which is a fully transformer-based ICR method. This is a substantially different architecture from the one in Table 1 and VSRN, and such transformer network architectures are the most prominent network architectures these days for multi-modal tasks (Radford et al., 2021; Chen et al., 2020d; Lu et al., 2019; Yuan et al., 2021). When only a limited amount of training data is available and one wants to make use of (partly) pre-trained transformer networks for multi-modal contrastive learning, constraint-based LTD can help to significantly improve the evaluation scores for ICR.

Furthermore, TERN makes use of a pre-trained BERT (Devlin et al., 2018) model as a caption encoder. BERT is a general-purpose text encoder pre-trained on text only. An open question is still why to train a caption encoder, while we use a (frozen) general-purpose sentence encoder to generate the latent targets for LTD; why not use the target decoder directly as a caption encoder? The results in Table 3 show that fine-tuning a general-purpose language encoder (i.e., BERT) with a contrastive loss as caption encoder results in lower evaluation scores than fine-tuning the caption encoder in combination with constraint-based LTD. This shows that LTD helps to extract features (i.e., not suppressing these features) from the input data that are relevant for the ICR task, that are not captured by either the pre-trained BERT model or by only using the contrastive optimization objective. In Appendix C, we provide three qualitative ranking results for i2t retrieval using TERN and samples from the COCO test set. Based on the examples it is clear that a baseline TERN does not represent specific concepts (i.e., predictive features) that are needed to rank the

| | | | i2t | | | t2i | | | | |
| | | | R@k | | | R@k | | | nDCG | |
| # | Method | Loss | 1 | 5 | 10 | 1 | 5 | 10 | ROUGE-L | SPICE |
|---|---|---|---|---|---|---|---|---|---|---|
| | | | **VSRN** | | | | | | | |
| | | | | | | F30k | | | | |
| 5.1.1 | BL | $\mathcal{L}_{con}$ | 60.3 | 85.6 | 90.8 | 44.9 | 74.0 | 83.3 | - | - |
| 5.1.2 | BL + LTD | $\mathcal{L}_{dual}, \beta = 1$ | 59.6 | 86.6 | 91.6 | 44.3 | 74.7 | 83.5 | - | - |
| 5.1.3 | BL + LTD | $\mathcal{L}_{lag}, \eta = 0.25$ | **61.9** | **86.8** | **92.1** | **45.3** | **76.4** | **84.4** | - | - |
| | | | | | | COCO | | | | |
| 5.2.1 | BL | $\mathcal{L}_{con}$ | 47.0 | 76.9 | 87.0 | 34.5 | 65.7 | 78.1 | 0.6779 | 0.6080 |
| 5.2.2 | BL + LTD | $\mathcal{L}_{dual}, \beta = 1$ | 45.9 | 77.8 | 87.6 | 34.6 | 66.2 | 78.3 | 0.6791 | 0.6088 |
| 5.2.3 | BL + LTD | $\mathcal{L}_{lag}, \eta = 0.15$ | **47.6** | **79.0** | **87.8** | **35.0** | **66.7** | **78.9** | **0.6797** | **0.6112** |
| | | | **TERN** | | | | | | | |
| | | | | | | COCO | | | | |
| 5.3.1 | BL | $\mathcal{L}_{con}$ | 41.2 | 72.6 | 83.6 | 31.0 | 61.9 | 74.7 | 0.6648 | 0.5926 |
| 5.3.2 | BL + LTD | $\mathcal{L}_{dual}, \beta = 1$ | 42.3 | 74.3 | 84.4 | 31.4 | 62.7 | 75.4 | 0.6684 | 0.5993 |
| 5.3.3 | BL + LTD | $\mathcal{L}_{lag}, \eta = 0.2$ | **44.1** | **74.8** | **85.7** | **33.6** | **64.6** | **76.9** | **0.6727** | **0.6059** |

Table 3: Recall@$k$ and nDCG evaluation scores for the F30k and COCO datasets using the VSRN and TERN network architectures. Boldface indicates the highest value for each evaluation metric per dataset. '-' indicates that it is not possible to compute the evaluation score for that dataset/experiment since the annotations are not available.

correct captions on top of the ranking, while TERN optimized with constraint-based LTD does represent these predictive features.

## 6 Conclusion

We have presented latent target decoding, a novel approach to reduce predictive feature suppression for contrastive resource-constrained ICR methods. Instead of reconstructing the captions in the input space, LTD reduces predictive feature suppression by reconstructing the input caption in the latent space of a general-purpose sentence encoder. By reconstructing the input caption, it is more difficult for the image and caption encoder to suppress predictive features that are not needed to solve the contrastive optimization objective.

**Main findings.** Our results show that constraint-based LTD obtains higher evaluation scores than both a contrastive ICR baseline and LTD implemented as a dual loss. This implies that we are able to reduce predictive feature suppression (and hence improve evaluation performance) by using constraint-based LTD, which does not require additional image-text training data or hard-negative mining strategies. Furthermore, we show that constraint-based LTD consistently results in a bigger improvement in evaluation scores than implementing LTD as a dual loss. These results suggest that, instead of simply minimizing both the contrastive and reconstruction loss, better evaluation scores can be obtained by only optimizing the reconstruction loss until a certain bound value $\eta$ is met. Finally, we show that constraint-based LTD can be combined with different contrastive learning losses and a wide variety of resource-constrained ICR methods.

**Implications.** The results of our work show that in a resource-constrained setup the evaluation performances of contrastive ICR methods can be substantially improved by using constraint-based LTD, without relying on more training data or hard-negative mining strategies. We, therefore, argue that, in a resource-constrained setup, LTD should be part of the standard ICR framework to mitigate the problem of predictive feature

suppression. Furthermore, we argue that when one uses an additional reconstruction objective to reduce predictive feature suppression, this objective should be considered to be implemented as an optimization constraint instead of a dual loss.

**Limitations.** In this work, we use a general-purpose sentence encoder to generate our latent target representation $\boldsymbol{y}_{\mathcal{C}k}$. However, we need to assume that this latent target representation contains the relevant information of the input caption. Furthermore, the availability of a general-purpose sentence encoder is not always guaranteed (e.g., when working with low-resource languages).

For the ICR task, the predictive features are the features needed to retrieve the positive item from a set of candidates. We, therefore, measure the reduction of predictive feature suppression by using the standard ranking evaluation metrics, such as recall@$k$, r-precision, and nDCG. However, we do not explicitly know which features are causing the observed improvement in the evaluation scores by using LTD.

**Future work.** We have several directions for future work. First, we plan to examine if the choice of different target generators will result in different ICR evaluation scores. Moreover, we also want to look into generating latent target representations without relying on a pre-trained sentence encoder.

Another promising direction for future work is analyses on the exact role of the optimization constraint. In Section 5.3 we examine the role of the optimization constraint when training the image and caption encoder. When the optimization constraint $\eta$ is met, $\lambda$ (i.e., the balancing parameter of the two losses) drops to zero and the reconstruction loss no longer provides a gradient (Figure 3b). Although $\lambda$ is (close to) zero for the majority of the training after the constraint is met, the evaluation scores on the validation set remain higher than when optimizing with $\mathcal{L}_{dual}$, with $\beta = 1$ (Figure 3a). This suggests that a constant gradient from the reconstruction loss does not benefit the training process, which is the case if LTD is implemented as a dual loss. We plan future research into the role of the optimization constraint, by trying constraint-based optimization for other multi-task optimization problems.

In this work, we focus on the reduction of predictive feature suppression for resource-constrained ICR methods. In Section 1 we argued that predictive feature suppression is less of an issue for models that are trained with large batch sizes since more information is needed to match the query with the positive candidate (due to a large number of negative candidates). However, it remains a promising direction for further research to investigate if and how constraint-based LTD can be used for either large-scale contrastive image-text representation learning or for fine-tuning. Prominent large-scale image-text matching methods, such as ALIGN (Jia et al., 2021), use noisy image-text pairs scraped from the internet. It is unclear if the target generator will provide useful targets (and hence a training signal) when the caption has a weak relation with its matching image (which is possible for noisy image-text pair). It might be the case that the target generator mainly provides useful supervision signals when using high-quality human-curated datasets, such as F30k and COCO.

Finally, we suggest working on methods to measure which features are responsible for the gained improvement in evaluation performance. A logical choice would be to use feature attribution methods. However, different feature attribution methods tend to disagree with each other, for both RNN and transformer-based models (Neely et al., 2021). Therefore, the choice of feature attribution method will influence the analyses of which predictive features are better captured by using LTD. To gain further insights into which features are captured by the learned encoders, we recommend developing task-specific feature attribution methods that can measure the reduction of predictive feature suppression directly.

## Acknowledgments

We thank the reviewers and the action editor for their valuable comments and suggestions. We thank Maartje ter Hoeve for reviewing the text. This research was supported by the Nationale Politie and the Hybrid Intelligence Center through the Netherlands Organisation for Scientific Research. All content represents the opinion of the authors, which is not necessarily shared or endorsed by their respective employers and/or sponsors.

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

# A   Notation and variables

| Symbol | Explanation |
|---|---|
| $\mathcal{L}_{con}$ | Symbol for the contrastive loss. In this work we either use the InfoNCE (van den Oord et al., 2018) or a triplet loss with in-batch hard-negative mining (Faghri et al., 2018). |
| $\mathcal{L}_{rec}$ | Symbol for the reconstruction loss of the input caption (i.e., *decoding loss*). In this work we use the negative cosine similarity when using embeddings (*latent target decoding*) or the log-likelihood when we reconstruct the tokens of the input captions (*input target decoding*). |
| $\mathcal{L}_{dual}$ | Symbol for the sum of the contrastive loss and the reconstruction loss (i.e., the dual loss). The reconstruction loss is scaled by $\beta$. |
| $\mathcal{L}_{lag}$ | Symbol for the sum of the contrastive loss and the reconstruction loss, where the reconstruction loss is implemented as a Lagrange multiplier optimization constraint. |
| $\boldsymbol{q}$ | Vector representation of a query, either an image or a caption. |
| $\boldsymbol{v}, \boldsymbol{v}^+, \boldsymbol{v}^-$ | Vector representation of a candidate. Given a query $\boldsymbol{q}$, a candidate is either matching ($\boldsymbol{v}^+$) or not matching ($\boldsymbol{v}^-$). Candidates are either images or captions. |
| $\mathcal{D}$ | Dataset consisting of $N$ image-caption tuples; each image $i \in N$ comes with $k$ captions. |
| $\boldsymbol{x}_{\mathcal{I}}^i$ | Input image $i$. |
| $\boldsymbol{x}_{\mathcal{C}_j}^i$ | Input caption $j$ that describes image $i$. |
| $\boldsymbol{z}_{\mathcal{I}}^i$ | Latent representation of image $i$. |
| $\boldsymbol{z}_{\mathcal{C}_j}^i$ | Latent representation of caption $j$ that describes image $i$. |
| $\eta$ | Reconstruction bound (or threshold). The reconstruction loss is only minimized up to the value of $\eta$. |
| $\lambda$ | The Lagrange multiplier. |
| $\beta$ | Balancing parameter to balance (or scale) two losses when using the dual loss. |
| $\mathcal{B}$ | Batch with training samples. |
| $\mathcal{S}_q^-$ | The set of all negative candidates $\boldsymbol{v}^-$, in a training batch, given query $\boldsymbol{q}$. |
| $\tau$ | Temperature parameter to scale the logist (i.e., cosine similarity) for the InfoNCE loss. |
| $\alpha$ | Margin parameter for the triplet loss. |
| $\boldsymbol{y}_{\mathcal{C}j}^i$ | Latent target representation (i.e., semantic embedding produced by a Sentence-BERT/language encoder) for caption $j$ that describes image $i$. |
| $\widetilde{\boldsymbol{y}}_{\mathcal{C}j}^i$ | Reconstructed latent target representation by the decoder network (i.e. LTD), for caption $j$ that describes image $i$. |
| $\widetilde{\boldsymbol{x}}_{\mathcal{C}j}^i$ | Reconstruction of the input tokens (i.e. ITD) by the decoder network for caption $j$ that describes image $i$. |
| $f_\theta(\cdot)$ | Image encoder parameterized by $\theta$. Takes as input $\boldsymbol{x}_{\mathcal{I}}^i$. Outputs a latent (global) image representation $\boldsymbol{z}_{\mathcal{I}}^i$. |
| $g_\phi(\cdot)$ | Caption encoder parameterized by $\phi$. Takes as input $\boldsymbol{x}_{\mathcal{C}j}^i$. Outputs a (global) latent caption representation $\boldsymbol{z}_{\mathcal{C}k}^i$. |
| $h_\omega(\cdot)$ | Decoder network parameterized by $\omega$. Takes as input $\boldsymbol{z}_{\mathcal{C}j}^i$. Outputs a reconstuction of the input caption, either $\widetilde{\boldsymbol{y}}_{\mathcal{C}j}^i$ (LTD) or $\widetilde{\boldsymbol{x}}_{\mathcal{C}j}^i$ (ITD). |

Table 4: Overview of the notation and variables used throughout this work.

## B  Gradient of the InfoNCE loss w.r.t. the query and candidates

We start with the definition of the InfoNCE loss (van den Oord et al., 2018) using the notation introduced in Section 3, for one query candidate pair $(\boldsymbol{q}, \boldsymbol{v}^+)$ and a set of negative candidates $(\mathcal{S}_{\boldsymbol{q}})$:

$$\mathcal{L}_{con} = -\log \frac{\exp(\boldsymbol{q}^T\boldsymbol{v}^+/\tau)}{\exp(\boldsymbol{q}^T\boldsymbol{v}^+/\tau) + \sum_{\boldsymbol{v}^-\in\mathcal{S}_{\boldsymbol{q}}^-}\exp(\boldsymbol{q}^T\boldsymbol{v}^-/\tau)} \tag{10a}$$

$$= -\left(\boldsymbol{q}^T\boldsymbol{v}^+/\tau - \log\left(\exp(\boldsymbol{q}^T\boldsymbol{v}^+/\tau) - \sum_{\boldsymbol{v}^-\in\mathcal{S}_{\boldsymbol{q}}}\exp(\boldsymbol{q}^T\boldsymbol{v}^-/\tau)\right)\right) \tag{10b}$$

$$-\mathcal{L}_{con} = \left(\boldsymbol{q}^T\boldsymbol{v}^+/\tau - \log\left(\exp(\boldsymbol{q}^T\boldsymbol{v}^+/\tau) - \sum_{\boldsymbol{v}^-\in\mathcal{S}_{\boldsymbol{q}}}\exp(\boldsymbol{q}^T\boldsymbol{v}^-/\tau)\right)\right). \tag{10c}$$

Next, we take the derivative of $-\mathcal{L}_{con}$ w.r.t. $\boldsymbol{q}$ (as also provided in (Chen et al., 2020c)):

$$-\frac{\partial\mathcal{L}_{con}}{\partial\boldsymbol{q}} = \boldsymbol{v}^+/\tau - \left(\exp(\boldsymbol{q}^T\boldsymbol{v}^+/\tau) - \sum_{\boldsymbol{v}^-\in\mathcal{S}_{\boldsymbol{q}}}\exp(\boldsymbol{q}^T\boldsymbol{v}^-/\tau)\right)^{-1}\cdot \tag{11a}$$

$$\left(\exp(\boldsymbol{q}^T\boldsymbol{v}^+/\tau)\boldsymbol{v}^+/\tau - \sum_{\boldsymbol{v}^-\in\mathcal{S}_{\boldsymbol{q}}}\exp(\boldsymbol{q}^T\boldsymbol{v}^-/\tau)\boldsymbol{v}^-/\tau\right)$$

$$= \boldsymbol{v}^+/\tau - \left(\frac{\exp(\boldsymbol{q}^T\boldsymbol{v}^+/\tau)}{\exp(\boldsymbol{q}^T\boldsymbol{v}^+/\tau) - \sum_{\boldsymbol{v}^-\in\mathcal{S}_{\boldsymbol{q}}}\exp(\boldsymbol{q}^T\boldsymbol{v}^-/\tau)}\right)\boldsymbol{v}^+/\tau - \tag{11b}$$

$$\sum_{\boldsymbol{v}^-\in\mathcal{S}_{\boldsymbol{q}}}\left(\frac{exp(\boldsymbol{q}^T\boldsymbol{v}^-/\tau)}{\exp(\boldsymbol{q}^T\boldsymbol{v}^+/\tau) - \sum_{\boldsymbol{v}^-\in\mathcal{S}_{\boldsymbol{q}}}\exp(\boldsymbol{q}^T\boldsymbol{v}^-/\tau)}\right)\boldsymbol{v}^-/\tau.$$

Now let us define $Z(\boldsymbol{q}, \boldsymbol{v})$ (similar to Eq. 2a in Section 3):

$$Z(\boldsymbol{q}, \boldsymbol{v}) = \frac{\exp(\boldsymbol{q}^T\boldsymbol{v}/\tau)}{\exp(\boldsymbol{q}^T\boldsymbol{v}^+/\tau) + \sum_{\boldsymbol{v}^-\in\mathcal{S}_{\boldsymbol{q}}^-}\exp(\boldsymbol{q}^T\boldsymbol{v}^-/\tau)}. \tag{12}$$

Next, we plug-in $Z(\boldsymbol{q}, \boldsymbol{v})$ into Eq. 11:

$$-\frac{\partial\mathcal{L}_{con}}{\partial\boldsymbol{q}} = \boldsymbol{v}^+/\tau - Z(\boldsymbol{q}, \boldsymbol{v}^+)\boldsymbol{v}^+/\tau - \sum_{\boldsymbol{v}^-\in\mathcal{S}_{\boldsymbol{q}}}Z(\boldsymbol{q}, \boldsymbol{v}^-)\boldsymbol{v}^-/\tau \tag{13a}$$

$$-\frac{\partial\mathcal{L}_{con}}{\partial\boldsymbol{q}}\tau = \boldsymbol{v}^+ - Z(\boldsymbol{q}, \boldsymbol{v}^+)\boldsymbol{v}^+ - \sum_{\boldsymbol{v}^-\in\mathcal{S}_{\boldsymbol{q}}}Z(\boldsymbol{q}, \boldsymbol{v}^-)\boldsymbol{v}^- \tag{13b}$$

$$= \left(1 - Z(\boldsymbol{q}, \boldsymbol{v}^+)\right)\boldsymbol{v}^+ - \sum_{\boldsymbol{v}^-\in\mathcal{S}_{\boldsymbol{q}}}Z(\boldsymbol{q}, \boldsymbol{v}^-)\boldsymbol{v}^-. \tag{13c}$$

In a similar way, we can take the derivative of $-\mathcal{L}_{con}$ w.r.t. $\boldsymbol{v}^+$ and $\boldsymbol{v}^-$:

$$-\frac{\partial\mathcal{L}_{con}}{\partial\boldsymbol{v}^+} = \boldsymbol{q}/\tau - Z(\boldsymbol{q}, \boldsymbol{v}^+)\boldsymbol{q}/\tau \tag{14a}$$

$$-\frac{\partial\mathcal{L}_{con}}{\partial\boldsymbol{v}^+}\tau = \left(1 - Z(\boldsymbol{q}, \boldsymbol{v}^+)\right)\boldsymbol{q}. \tag{14b}$$

$$-\frac{\partial\mathcal{L}_{con}}{\partial\boldsymbol{v}^-} = -Z(\boldsymbol{q}, \boldsymbol{v}^-)\boldsymbol{q}/\tau \tag{15a}$$

$$-\frac{\partial\mathcal{L}_{con}}{\partial\boldsymbol{v}^+}\tau = -Z(\boldsymbol{q}, \boldsymbol{v}^-)\boldsymbol{q}. \tag{15b}$$

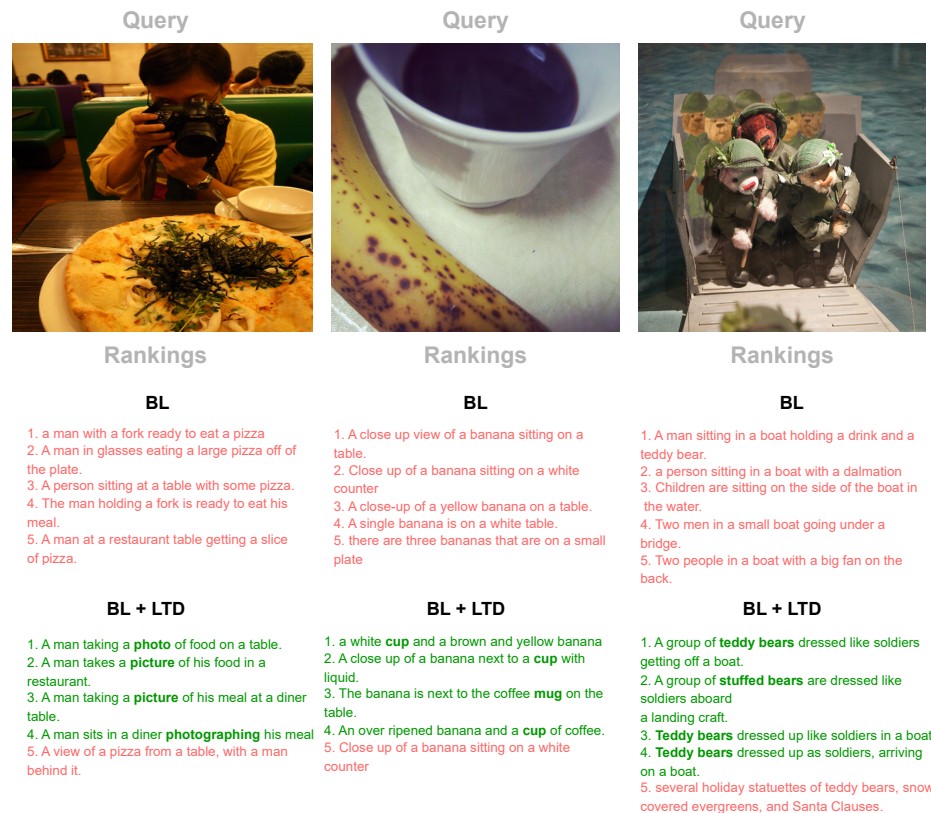

Figure 4: Three query images from the COCO test set. For each image query we show the top 5 retrieved captions by TERN. We compare the TERN baseline (BL) and TERN optimized in combination with constraint-based LTD (BL + LTD). Ground-truth captions (i.e., matching) are indicated in green. Captions in red indicate captions that do not match with the query image.

## C  Ranking examples

In Figure 4 we provide three query images from the COCO test set and the top 5 retrieved captions by TERN. We compare the TERN baseline (BL) and TERN optimized in combination with constraint-based LTD (BL + LTD). We selected three examples with a large difference in precision@5 between the BL and BL + LTD.

For all three examples, it is clear that the baseline ICR methods miss a concept (i.e., *predictive feature(s)*) that is needed to rank the ground-truth captions in the top 5. In the left example, the best matching captions according to the BL ignore that the man in the image *takes a photo*. In the middle example, the best matching captions, according to the BL, do not match on the fact that there is a *cup/mug*. In the right example, the best matching captions do not contain the concept of *teddy bears*. Clearly, an ICR method optimized in combination with LTD is able to match images and queries based on more fine-grained features in the images and captions than a baseline ICR method.

