# OpenReview forum: "Reducing Predictive Feature Suppression in Resource-Constrained Contrastive Image-Caption Retrieval"
_TMLR — Accepted by TMLR_

### Review · Reviewer_zvm2 · 2023-01-10

**Summary Of Contributions:**

This paper proposes latent target decoding (LTD) to reduce predictive feature suppression for resource-constrained image-caption retrieval (ICR) methods. An decoder is added to the contrastive ICR framework for the reconstruction of the input caption in a latent space to prevent the image and caption encoder from suppressing predictive features. The LTD objective is implemented as an optimization constraint to optimize the reconstruction loss as well as the contrastive loss. Experiments are conducted on two popular ICR benchmark datasets to show the effectiveness of the propopsed method.

**Audience:**

Yes

**Broader Impact Concerns:**

None.

**Claims And Evidence:**

Yes

**Requested Changes:**

Please see the weaknesses above to revise this paper.

**Strengths And Weaknesses:**

Strengths:
1. The studied toIs an image worth five sentences? a new look into semantics for image-text matchingpic, i.e. image-caption retrieval, is a popular research task to bridge the cross-modal gap. The motivation is clear and the idea to investigate resource-constrained image-caption retrieval is novel.
2. The proposed method, i.e., latent target decoding (LTD) is well presented and easy to follow. Genearlly, LTD is reasonable and makes good sense.
3. The reported results are superior to the compared baselines. Different aspects of the proposed method are analyzed in experiments.

Weankesses and suggestions:
1. The compared baselines are insufficient. There are many recent image-caption retrieval methods, such as Negative-Aware Attention Framework for Image-Text Matching", "Is an image worth five sentences? a new look into semantics for image-text matching", "Cross-Modal Image-Text Retrieval with Semantic Consistency". Introducing and comparing with them in both related work and experimental part is necessary.
2. Some detailed retrieval resutls are expected. Please give both successful and failed cases and analyze the reasons.
3. A visualized example would be better to show the motivation.
4. Therr are many notations and equations. A talbe summarizing the used variables will make it easier for users to understand.

---

### Review · Reviewer_ApZs · 2023-01-19

**Summary Of Contributions:**

The paper discusses the Image-Caption retrieval (ICR) task. Commonly the task is solved with contrastive learning. This approach might pick on a few "predictive features" (i.e., that allow discriminative learning between positive and negative pairs) during training. The authors propose to have an additional decoder head to reconstruct the input caption embedding (namely, latent target decoding).

Main ideas:
i) reconstruct the latent space of the caption instead of the caption directly.
ii) use the reconstruction as a constraint instead of a dual optimization

Evaluation is done on image-to-text and text-to-image tasks on the COCO, COCO CxC, and F30k datasets with two different architectures, VSRN and TERN.

**Audience:**

Yes

**Broader Impact Concerns:**

It may be interesting to study if predictive features can cause undesirable biases, and if this new suggestion fixes it or makes it worse.

**Claims And Evidence:**

No

**Requested Changes:**


Overall the idea is intresting, but the current results only show that the direction is promising, but it is not a thorough study. Changes should answer the weakness. Specifically, the authors should focus on defining the problem - with metrics that assess it and qualitative analysis. The improvements should be on both the proposed metrics and qualitatively. An intresting analysis could be employing this method to large-scale models (e.g., CLIP) with fine-tuning.


Further, it might be a matter of styling, but the writing can be more concise, e.g., these two seances repeating the same thing.
"Implementing LTD as dual loss, as opposed to an optimization constraint, does not reduce predictive feature suppression. Our analyses suggest that optimizing the reconstruction loss only until a specific bound value is met results in better evaluation performance than minimizing the reconstruction loss as a dual loss." Also, there are some typos, like the word `viz' at the end of page 2.




**Strengths And Weaknesses:**

Strengths:

(+) The idea is interesting. Contrastive models tend to focus on a low number of discriminative features  (Robinson et al. 2021). Pushing the representation to keep relevant information for reconstruction should help it avoid relying only on discriminative features.
(+) The paper suggests an important design choice. The reconstruction loss should require similarity to latent space (of the caption) instead of decoding the tokens directly.
(+) The authors show two variations for the regularization. That is as a dual loss (calibrated with hyperparameter) and constraint-based that does not require intensive tuning and can instead optimize a multiplier.
(+) The related work section is comprehensive.




Weaknesses:
The evidence that the method does with it intends to do, i.e.,  reducing predictive feature suppression, needs to be stronger.

(W1) Retrieval Metric improvement is not proving cause. The authors did not show any evidence that the reason for the improvement is related to "reducing feature suppression": There is no analysis that shows the problem exists. The paper should be more self-contained, and here is no qualitative analysis or metrics related to this specific issue.

(W2) There is no comparison to baseline results. The paper only compares against re-implementations. It limits the ability to get a real sense of how robust the method is. Why not follow the evaluation setups presented by Robinson et al. 2021?

(W3) The resource-constrained setup is not defined or justified. Since the method can be applied to existing architectures, the authors can also fine-tune existing large-scale solutions employing their new loss. Such analysis is not provided.

(W4) In general, contrastive loss has been found to be an efficient way to acquire zero-shot capabilities by training on large-scale data (Radford et al., 2021). It is intresting to show if the new loss idea can improve this capability. It is especially intresting since Radford et al., 2021 argued that they could not achieve the same zero-shot capabilities in the reconstruction setup.

---

### Review · Reviewer_W1M4 · 2023-01-29

**Summary Of Contributions:**

This work proposes latent target decoding, a method to improve representations learnt through pretraining with contrastive learning. Typical contrastive losses do not enforce constraints that require learnt representations to contain all available information. LTD is motivated as an optimization constraint which measures the reconstruction quality of an input caption in the latent space of a language encoder. The paper shows the positive effect of LTD on various contrastive learning setups for image-text retrieval.

**Audience:**

Yes

**Broader Impact Concerns:**

There are no major ethical or impact concerns introduced in this paper. This work is about contrastive training to learn features, and the datasets used are fairly standard in the community.

**Claims And Evidence:**

Yes

**Requested Changes:**

## Critical for acceptance:
- Show experiments with more competitive contrastive learning setups such as CLIP
- Include experiments showing the effect of LTD over different batch sizes. It is important to know that the proposed method will still be helpful at scale, and its effects will not be washed out with larger batch sizes (which many SOTA models use) and larger model sizes.


## Would strengthen the work:
- Provide the intermediate steps for derivation of the gradients in eq 2b, 2c, and 2d
- Show results with sentence/language encoders other than Sentence-BERT
- Include hyperparameter details into the main text. The authors mention that it is the same as Chun et al., 2021, but certain details (such as batch size) are not found in Chun et al., 2021 either.



**Strengths And Weaknesses:**

The paper is well written, and the proposed approach is well motivated and easy to follow. I appreciate the contributions, but hold some concerns with the experimental results. I believe that the current experiment findings do not comprehensively justify the benefits of LTD over vanilla contrastive learning.

## Strengths:
- The proposed approach LTD is well motivated. The paper explains in detail why feature suppression is an unintended consequence of previous contrastive learning frameworks, and why other methods (such as autoencoding) are insufficient.
- The proposed approach is implemented as an optimization constraint, which minimizes hyperparameter tuning.
- LTD is general: it works for both InfoNCE and triplet loss, which most contrastive pretraining methods use.


## Weaknesses:
- I am not sure that the derivations presented in eq 2b, 2c and 2d are correct. I get a slightly different result when computing it. Can you please provide details for the derivatives?
- It’s suggested in the paper that LTD is agnostic to the choice of the language encoder. Could you show some experiments with other pretrained encoders, and show how the results vary with different models?
- The architecture used is not competitive with most SOTA contrastive learning methods. In particular, the caption encoder is a single layer GRU, which seems to be significantly worse off in general compared to the larger transformers used in CLIP and ALIGN. It’s unclear if LTD will generalize to stronger contrastive learning setups.
- The generalization experiments are run on VSRN and TERN, which are fine. However, I’m not sure why a CLIP-like model was not used instead, since that appears to be one of the stronger open sourced baselines used in many contrastive learning tasks.
- The batch size used in experiments seems to be 128 (from the anonymized code). As mentioned by the authors, most SOTA contrastive learning approaches benefit from large batch sizes (for example, CLIP uses a batch size of 32768). Can you show some form of scaling curve as batch size increases, to motivate that LTD will still be beneficial at larger scales?


## Minor issues
- “Upshot” (page 4 and 6) is a strange (and I believe non-standard) way to phrase the conclusion. Consider removing this header.
- Missing space on page 8: “(Robinson et al., 2021)and”

---

> ### Author Response · Authors · 2023-02-03
> **Response to Reviewer W1M4**
>
> We would like to thank the reviewer for the review and the encouraging suggestions. We reply to each weakness individually.
>
> - **I am not sure that the derivations presented in eq 2b, 2c and 2d are correct.  Can you please provide details for the derivatives?**
>
> We want to thank the reviewer for the suggestion and we added the full derivative to the appendix. We use the derivative provided by [1] (Table 2), but changed the notation for our task. We take the derivative of $- L_{con}$ (and not $ L_{con} $), and we moved the temperature parameter $\tau$ to the other side of the equation (since it scales the gradient). We realized that we made a slight notation mistake in the denominator of $Z(q, v)$, which we solved. However, this does not change the gradient in 2b, 2c, and 2d. We hope that this clarifies the derivative.
>
> - **It’s suggested in the paper that LTD is agnostic to the choice of the language encoder. Could you show some experiments with other pretrained encoders**
>
> We want to thank the reviewer for suggesting the experiment. Indeed, LTD is agnostic to the choice of language encoder. We have a preliminary experiment  (not provided in the paper) where we used a version of the SentenceBert model [2] for the target encoder that differs from the version used in Section 5.1 (in the paper we use [3] as target encoder). [2] provides smaller target embeddings (384-d instead of  768-d) than [3]. We provide a subset of the results in the table below.
>
> | dataset | method   | loss | R@1      | R@5      | R@10     | R@1      | R@5      | R@10     |
> |---------|----------|------|----------|----------|----------|----------|----------|----------|
> | F30k    | BL       | con  | 45.0     | 76.4     | 85.5     | 33.2     | 65.5     | 76.3     |
> | F30k    | BL + LTD | dual | 45.8     | 75.3     | 84.6     | 33.7     | 66.0     | 77.0     |
> | F30k    | BL + LTD | lag  | **49.7** | **78.9** | **86.7** | **37.5** | **68.6** | **79.5** |
> |---------|----------|------|----------|----------|----------|----------|----------|----------|
> | COCO    | BL       | con  | 32.9     | 64.3     | 76.5     | 23.8     | 53.3     | 66.9     |
> | COCO    | BL + LTD | dual | 34.4     | 64.1     | 76.7     | 24.5     | 54.0     | 67.5     |
> | COCO    | BL + LTD | lag  | **35.1** | **66.7** | **78.6** | **25.6** | **55.0** | **68.7** |
>
> Based on these results we can conclude that constrained-based LTD also outperforms the baseline when using a different target generator. (As an aside, the BL results are lower than in the current version of the paper. The reason for this is that these are older (very preliminary) experiments and we slightly improved our baseline for our final experiments.)
>
> - **The architecture used is not competitive with most SOTA contrastive learning methods. In particular, the caption encoder is a single layer GRU, which seems to be significantly worse off in general compared to the larger transformers used in CLIP and ALIGN. It’s unclear if LTD will generalize to stronger contrastive learning setups.**
>
> We respond to two aspects of the reviewers’ comment.
>
> **First, “..the caption encoder is a single-layer GRU, which seems to be significantly worse off in general compared to the larger transformers used in CLIP and ALIGN.”**
>
> Indeed, in Sections 5.1-5.4, we use a single-layer bidirectional GRU (similar to for example [4]). However, the main goal of these first experiments is to show that LTD works in small-scale training setups.
> Moreover, we also apply LTD in combination with TERN. TERN makes use of a BERT-based caption encoder, which consists of 6 transformer layers plus two additional transformer layers that are shared between the image and caption encoder.
> The caption encoder of CLIP consists of a 12-layer (GPT-like) transformer encoder (with a varying number of attention heads and widths). Although TERN does not use the same number of transformer layers for the caption encoder, we do use a transformer-based caption encoder in this work as well. CLIPs requirements to train from scratch are extreme compared to our resource-constrained setup.
>
> **Second, “It’s unclear if LTD will generalize to stronger contrastive learning setups.”**
>
> Indeed, it is unclear if LTD generalizes to very large scale-data setups, such as CLIP or ALIGN, and we do not make any claim w.r.t. the generalization of LTD to such setups. At some point, big models such as CLIP will have seen enough language data that a method like LTD should not add any value anymore. We expect LTD to stop helping at some larger model and/or data size. However, due to computational limitations, we are not able to empirically test this.
> In this work, we focus on resource-constrained setups, i.e., models that are only trained and evaluated on the COCO and F30k datasets. We agree that it is an interesting direction for future work to apply LTD in combination with large-scale training or fine-tuning for future work.
>
> (continues in the next comment)

---

> > ### Author Response · Authors · 2023-02-03
> > **Response to Reviewer W1M4**
> >
> > - **The generalization experiments are run on VSRN and TERN, which are fine. However, I’m not sure why a CLIP-like model was not used instead, since that appears to be one of the stronger open sourced baselines used in many contrastive learning tasks.**
> >
> > As argued in the paper, we focus on models that are both trained and evaluated on the F30k and COCO datasets. Indeed, CLIP is a stronger open-source baseline but does not fit in that setup since it is pre-trained on 400 million image-text pairs and evaluated in a zero-shot fashion for ICR.  We are not able to reproduce CLIP in combination with LTD due to computational constraints. We do think that it is an interesting direction for future work to explore the use of LTD either in combination with large-scale image-text pre-training or for fine-tuning large-scale image-text models for a specific task. Another interesting future direction would be to use LTD as a fine-tuning strategy. We have added this to Section 6 of the revision of our paper.
> >
> > - **The batch size used in experiments seems to be 128 (from the anonymized code). As mentioned by the authors, most SOTA contrastive learning approaches benefit from large batch sizes (for example, CLIP uses a batch size of 32768). Can you show some form of scaling curve as batch size increases, to motivate that LTD will still be beneficial at larger scales?**
> >
> > Thanks for the question. To clarify, all experiments make use of a batch size of 128. We added this to a revised version of the paper. Many works [4, 6,7,8] that focus on ICR, and that only train on F30k and COCO, use a batch size of 128. To stay consistent with previous work we also use a batch size of 128.
> > We want to thank the reviewer for suggesting the batch-size scaling experiment. F30k consists of 29k training images and MS-COCO roughly 113k. Moreover, all models we train and evaluate in this work run on a single GPU. Hence, we are not able to get even close to the extreme batch size of models such as CLIP.
> >
> > However, we do think that a batch-size scaling experiment is a valuable contribution to the paper and shows the strength of LTD in small-scale training setups. Therefore, we rerun the experiment from Section 5.1 (both for F30k and COCO) with a batch size of {4, 8, 16, 32, 64, 256}. We use the recall sum (the sum of R@{1,5,10} for both i2t and t2i) as an evaluation metric, to get an indication of the total recall scores. We provide the scores for the contrastive baseline and for LTD with $L_{lag}$ (with a similar value for $\eta$ as used in Section 5.1).
> > We do not have all the results yet, and we will share them as soon as we have them. However, based on the intermediate results for F30k we can already draw a general conclusion: constraint-based LTD always outperforms a contrastive baseline with a good margin, regardless of the batch size.
> >
> > | Batch size | baseline | baseline + LTD |
> > |---|---|---|
> > | 4 | 293.86 | 334.18 |
> > | 8 | 359.13 | 395.22 |
> > | 16 | 381.18 | **409.54** |
> > | 32 | **388.28** | 406.23 |
> > | 64 | 378.42 | 403.80 |
> > | 128 | 379.48 | 398.16 |
> > | 256 | 371.69 | 398.89 |
> >
> > It is interesting to observe that a bigger batch size does not always results in higher recall for the baseline (a similar trend can be found in Figure 9 from [1]). However, It is important to mention that when the batch size is doubled, the number of training iterations, and hence the number of model updates, is reduced by 50%. Therefore, a model with a higher batch size should also train twice as long to get a fair number of updates.
> >
> > ### References
> >
> > [1] A Simple Framework for Contrastive Learning of Visual Representations. Ting Chen, Simon Kornblith, Mohammad Norouzi, Geoffrey Hinton.
> >
> > [2] https://huggingface.co/sentence- transformers/all- MiniLM- L6- v2
> >
> > [3] https://huggingface.co/sentence-transformers/all-mpnet-base-v2
> >
> > [4] Probabilistic Embeddings for Cross-Modal Retrieval. Sanghyuk Chun, Seong Joon Oh,Rafael Sampaio de Rezende, Yannis Kalantidis, Diane Larlus.
> >
> > [5] Scaling Up Visual and Vision-Language Representation Learning With Noisy Text Supervision. Chao Jia, Yinfei Yang, Ye Xia, Yi-Ting Chen, Zarana Parekh, Hieu Pham, Quoc V. Le, Yunhsuan Sung, Zhen Li, Tom Duerig
> >
> > [6] Is An Image Worth Five Sentences? A New Look into Semantics for Image-Text Matching. Ali Furkan Biten, Andres Mafla, Lluis Gomez, Dimosthenis Karatzas.
> >
> > [7] VSE++: Improving Visual-Semantic Embeddings with Hard Negatives. Fartash Faghri, David J. Fleet, Jamie Ryan Kiros and Sanja Fidler.
> >
> > [8] Visual Semantic Reasoning for Image-Text Matching. Kunpeng Li, Yulun Zhang, Kai Li, Yuanyuan Li and Yun Fu.

---

> > > ### Author Response · Authors · 2023-03-07
> > > **Follow up on the batch-scaling experiments**
> > >
> > > We want to follow up on the batch-scaling experiments we provided in our previous comment.
> > > The previous comment only provides the batch-scaling results for the F30k dataset.
> > > However, we have also conducted the same experiments for COCO.
> > > We omit the experiment with a batch size of 4 and 8 - training the model with a batch size of 4/8 for 60 epochs on the entire COCO dataset simply takes too long.
> > > Based on the table with results included below, we can draw the same conclusion for COCO as for F30K: constraint-based LTD consistently outperforms a contrastive baseline with a good margin, regardless of the batch size.
> > >
> > >
> > >
> > >
> > > Recall sum, for the COCO validation set, measured for models trained with different batch sizes.
> > > | Batch size  | Baseline | Baseline + LTD |
> > > |-------------|----------|----------------|
> > > | 4           | -        | -              |
> > > | 8           | -        | -              |
> > > | 16          | 297.33   | **311.38**     |
> > > | 32          | 316.93   | **332.73**     |
> > > | 64          | 321.57   | **335.18**     |
> > > | 128         | 321.25   | **335.08**     |
> > > | 256         | 320.76   | **334.85**     |

---

### Author Response · Authors · 2023-02-07
**Uploaded revision of the paper**

We would like to thank all the reviewers for their reviews and constructive feedback. We have uploaded a revised version of the paper. All changes in the text are highlighted in green.  The main points addressed in the paper are:

- To better motivate the problem, we added a visualized example of predictive feature suppression in small-scale contrastive learning setups to the Introduction (Reviewer **zvm2**).
- We slightly changed the definition of 'predictive feature suppression for ICR' in the Introduction.
- We added the related work suggested by Reviewer zvm2 to Section 2.
- We added a table with all the symbols and variables used throughout the paper to the Appendix (Reviewer **zvm2**).
- We added some visual examples of rankings (successes and failures) for a baseline ICR method and a baseline + LTD to the Appendix (Reviewer **zvm2**).
- We added the complete derivations for Eq. 2b, 2c, and 2d to the Appendix (Reviewer **W1M4**).
- We fixed a notation mistake in Eq. 2a.
- We added more details about the batch size and training details to Section 4 (experimental setup) (Reviewer **W1M4**).
- We better define the resource-constrained setup by explicitly stating that all models are only trained and evaluated on either the F30k or the COCO dataset, and trained on a single GPU.
- We added a paragraph to Section 6 discussing future work on applying LTD in combination with large-scale image-text pre-training or fine-tuning.
- We added a discussion on why the current formulation of LTD does not work with (contrastive) ICR methods that do not compute a global representation of both the image and the caption to Section 3.

We are happy to continue the discussion and to further address any questions or requests the reviewers might have.

---

### Decision · Action_Editors · 2023-03-15

**Recommendation:** Accept with minor revision

**Comment:**

While most issues raised during reviewing were addressed, several questions by ApZs were not incorporated or replied to. Minimally, there are some lingering concerns about how to evaluate if the proposed approach improves robustness (either additional evaluations or metrics).  zvm2 also requests some additional visualization of examples.  While additional experiments would be an ideal inclusion for ApZs, minimally, the authors should try to adjust claims or indicate places for future work to address the listed weaknesses/concerns.

**Audience:**

The vision-language literature and contrastive learning more broadly

**Claims And Evidence:**

A proposed change to contrastive learning -- introducing a latent target reconstruction loss -- to avoid features impression.  The approach is applied to image-caption retrieval and aims to prevent models from reducing/removing redundant predictive features. Experiments are performed on MS-COCO and Flicker30k and the retrieval results are convincing -- though CL has been applied to substantially larger domains so there is no direct comparison to state-of-the-art models in this space, this and further analysis of the role of the reconstruction targets are left to future work.

---

> ### Author Response · Authors · 2023-04-03
> **Reply on decision by Action Editors**
>
> Dear Reviewers and Action Editors(s),
>
> Thank you for your time and efforts in reviewing and accepting our paper.
>
> - We have provided the requested additional visualization of examples (reviewer **zvm2**) in Appendix C of the revised paper.
>
> - Reviewer ApZs raised the following concern: "There is no comparison to baseline results. The paper only compares against re-implementations. It limits the ability to get a real sense of how robust the method is. Why not follow the evaluation setups presented by Robinson et al. 2021?"
> As we state in our response, we do not use re-implementations, but the official code provided by the authors. We follow the setups presented by Robinson et al. 2021 (to the extent that they can be compared). We add LTD on top of the baseline and then compare baseline + LTD in a fair manner to a baseline without LTD; see Sections 4 and 5 of the revised paper. Thus, we believe that we provide the type of robustness that reviewer **ApZs** is asking for.
>
> Thank you again for your valuable feedback on our paper.
>
> The authors